# SAMerging: Sharpness-aware Model Merging via Multi-Teacher Knowledge Distillation

## Abstract

Model merging offers a lightweight alternative to joint multi-task learning (MTL), which is often costly or data-prohibitive. While the task arithmetic seems promising, it is brittle to coefficient scaling, and we observe that recent approaches, such as AdaMerging, that learn these coefficients, remain sensitive to initialization. This raises a key question: can merging coefficients be learned in a principled, label-free way? We introduce SAMerging, a method that learns coefficients by seeking flat minima. Our approach is grounded in two theoretical contributions. First, we derive a flatness-aware PAC-Bayes generalization bound for the merged model, featuring a novel cross-task heterogeneity term that quantifies expert-task mismatch. Second, this analysis guides us to frame merging as multi-teacher knowledge distillation on a small, unlabeled dataset. We formally show that minimizing the student-teacher KL divergence tightens an upper bound on the merged model's excess risk. We then employ Sharpness-Aware Minimization (SAM) to find robust solutions that generalize better. Empirically, SAMerging establishes a new state of the art on vision and NLP benchmarks. Notably, it surpasses AdaMerging with accuracy gains of $+4.5\%$ on TA-8 and $+11.7\%$ on TALL-20. This is achieved with remarkable data efficiency, using $10\times$ fewer calibration data and proving effective even in data-scarce settings with as few as 16 examples per task. Furthermore, it requires no original training data and incurs no additional inference-time or memory overhead.

## 1 Introduction

The pretrain-fine-tune paradigm has become the dominant approach for obtaining expert models for various tasks, including Natural Language Processing (NLP) and Computer Vision (CV). Recently, due to the increasing use of these models in resource-limited devices, there has been growing interest in developing models that can handle multiple tasks simultaneously. One line of research, namely model merging, leverages the existing fine-tuned models to achieve the multi-task model's parameters (Ilharco et al., 2023; Matena & Raffel, 2022; Wortsman et al., 2022). Model merging seeks to combine fine-tuned models into a single model that retains the specialized capabilities of each task-specific expert (Breiman, 1996; Chen & Guestrin, 2016; Ganaie et al., 2022), without the need to run multiple constituent models, so both inference cost and memory cost will be in $\mathcal{O}(1)$ instead of $\mathcal{O}(n)$ for $n$ models (Yang et al., 2024a). Moreover, due to limited data access, privacy concerns, and high fine-tuning costs, model merging is gaining interest, especially in privacy-preserving settings like federated learning (Tao et al., 2025; Liu et al., 2024b; Chen et al., 2025; Salami et al., 2025; Tsouvalas et al., 2025).

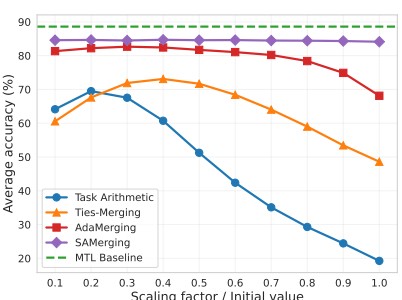

Figure 1: **Sensitivity to merge scaling and initialization.** On TA-8, we compare merge scaling (TA/TIES) and initialization (AdaMerging/SAMerging). While designed to learn coefficients, *AdaMerging*'s performance is sensitive to initialization, suggesting its objective/optimizer is a bottleneck. In contrast, SAMerging attains higher and more stable accuracy across the range.

One line of methods in model merging is based on the notion of "task arithmetic" (Ilharco et al., 2023), which treats each expert's offset from the pretrained weights as a task vector. Scaling and summing these vectors across tasks can yield a multi-task model with expert-level performance. This

insight has inspired numerous merging methods (Ortiz-Jimenez et al., 2023; Yadav et al., 2023; Yang et al., 2023) and theory on when it succeeds (Li et al., 2025; Zhou et al., 2024; Wang & Wang, 2024). However, performance is highly sensitive to the scaling coefficients (see Fig. 1), even for approaches like AdaMerging that aim to learn the merging coefficients (Yang et al., 2023). This motivates more principled ways to learn these coefficients for stronger generalization.

Flatter loss landscapes have long been associated with better generalization (Hochreiter & Schmidhuber, 1997; Neyshabur et al., 2017; Petzka et al., 2021; Andriushchenko et al., 2023; Yue et al., 2023; Haddouche et al., 2025), inspiring metrics and algorithms such as Sharpness-Aware Minimization (SAM), which explicitly seeks wider minima to improve generalization (Foret et al., 2021). In multi-task learning (MTL), classical theory attributes gains to shared representations and inductive biases (Caruana, 1997; Baxter, 2000; Argyriou et al., 2006; Maurer et al., 2016; Zhang & Yang, 2021; Zakerinia et al., 2025), yet heterogeneous tasks often suffer negative transfer and interference (Tsouvalas et al., 2025; Zakerinia & Lampert, 2025). These observations suggest that solution geometry, favoring flatter minima, can mitigate cross-task interference by reducing sensitivity to parameter perturbations and stabilizing shared features, thereby improving MTL generalization (Dai & Zhu, 2020; Dinh et al., 2017; Andriushchenko et al., 2023). It is thus natural to explore flatness-aware optimization (e.g., SAM) for MTL fine-tuning and merging; Lee et al. (2025) takes a step in this direction for fine-tuning.

We introduce a new perspective on model merging by deriving a PAC-Bayes generalization bound that directly connects the generalization of the merged model to the flatness of its loss basin. Inspired by the emerging links between flatness (Haddouche et al., 2025), multi-task learning (Zakerinia & Lampert, 2025), and AdaMerging (Yang et al., 2023), we propose `SAMerging`, a novel method that merges fine-tuned experts while explicitly seeking flatter solutions. Unlike previous approaches, `SAMerging` is designed to retain task-specific performance while promoting generalization through geometry-aware merging. It achieves state-of-the-art results across diverse tasks in CV and NLP and remains robust under variations in data size and task count. In summary, our contributions are:

- The central contribution of the present work is a new PAC-Bayes generalization bound for multi-task model merging, which formally captures the gap between two paradigms: merging independently fine-tuned models (*a mixture of models*) versus jointly training on *a mixture of data*. This bound introduces a novel discrepancy term that quantifies how merging diverges from standard multi-task learning, offering a principled lens through which to analyze and improve merging strategies.

- Building on this insight, we propose a new merging objective inspired by multi-teacher knowledge distillation. Specifically, we minimize the Kullback-Leibler (KL) divergence between the merged model and each task-specific expert to align the merged model's behavior with each expert on its respective task. This approach not only aligns with our theoretical analysis but also outperforms entropy-based merging criteria such as those used in AdaMerging (Yang et al., 2023).

- Finally, motivated by the PAC-Bayes connection between generalization and flatness, we incorporate sharpness-aware minimization into the merging process. This promotes flatter solutions during optimization, leading to further gains in performance. Our ablations show that the KL-based objective and SAM contribute complementary benefits, with their combination yielding state-of-the-art results across both vision and language tasks.

## 2 RELATED WORK

**Joint training for multi-task learning (MTL)**. Joint training for MTL aggregates data from different tasks to learn them jointly. This method enables knowledge transfer with inductive bias and shared representations (Caruana, 1997; Baxter, 2000; Wu et al., 2023). Prior work tries to tackle this problem by working either on the (i) architecture of the models or (ii) training and optimization regime. On the architecture, work includes refining cross-task coupling (Misra et al., 2016), selective sharing (Sun et al., 2020), and mixture-of-experts to learn which experts to share per task (Hazimeh et al., 2021; Tang et al., 2020). On the training and optimization, it focuses on mitigating gradient conflicts (Yu et al., 2020; Liu et al., 2024a; Quinton & Rey, 2025), adjusting training weights (Kendall et al., 2018; Chen et al., 2018), or formulating MTL as multi-objective optimization to seek Pareto-optimal trade-offs with convergence guarantees and controllable preferences (Lin et al., 2019; Shamsian et al.,

2023). Joint MTL can boost efficiency and generalization, but in the foundation-model era, it is often impractical to pool raw data and update large backbones due to compute and privacy constraints. Meanwhile, the rise of fine-tuned experts on platforms like Hugging Face motivates post-hoc *model merging*, shifting heterogeneity and task-interference challenges from data to models.

**Model merging for MTL**. In contrast to joint MTL, *model merging* fuses multiple task-specific fine-tuned experts into one MTL model (Yang et al., 2024a). Data-free approaches include simple weight averaging/model soups (Wortsman et al., 2022) and task arithmetic (TA) (Ilharco et al., 2023). TA has led to numerous new methods for merging data-free approaches, such as TIES-Merging (Yadav et al., 2023), DARE (Yu et al., 2024), PCBMerging (Du et al., 2024), and Isotropic Merging (Marczak et al., 2025). Beyond data-free merging, data-dependent methods use unlabeled samples per task for a one-time *offline* calibration. The deployed model remains a single network with no extra parameters. Concretely, Fisher Merging estimates Fisher information from gradients on unlabeled data to make the Fisher Information Matrix (FIM) (Matena & Raffel, 2022); RegMean/RegMean++ compute feature inner-product statistics to regularize averaging (Jin et al., 2025; Nguyen et al., 2025), and AdaMerging learns (layer-/task-wise) merge coefficients by minimizing predictive entropy (Yang et al., 2023). By contrast, methods that learn per-task heads/masks/adapters introduce inference-time compute and memory overhead (Yang et al., 2024b). In deployed settings, models already see inputs from target domains, so calibrating on a handful of unlabeled samples is a far weaker assumption than joint MTL's requirement for pooled training data. We thus trade minimal label-free calibration for *zero* inference overhead. We achieve this by demonstrating that `SAMerging` will reach state-of-the-art performance by utilizing as few as one batch of size 16 for each task. After calibration, one backbone serves all tasks with $\Theta(1)$ memory and latency, whereas ensembles, adapters, or per-task heads/masks incur runtime and memory costs (Yang et al., 2024b).

**Knowledge distillation**. We cast model merging as multi-teacher knowledge distillation (Hinton et al., 2015; Yang et al., 2023; Xu et al., 2025): compress an *ensemble of experts* into a single student by minimizing the KL divergence between the ensemble's soft predictive distribution and the student's outputs on unlabeled data, as in (Hinton et al., 2015). This function-space target is robust to weight misalignment and unit permutations. In contrast to element-wise, feature-level merging (Xu et al., 2025), we analyze the merged model's predictive distribution, which enables explicit excess-risk guarantees.

**Flatness, generalization, and MTL**. Extensive evidence links wider (flatter) minima to better generalization and robustness, and PAC-Bayesian analysis formalizes the link such that when weight posterior concentrates in a broad low-loss region, complexity terms shrink, yielding non-vacuous bounds (Neyshabur et al., 2017; Petzka et al., 2021; Dziugaite & Roy, 2017). SAM (Foret et al., 2021) achieves flatter minima by penalizing the worst-case loss in a neighborhood, thereby improving generalization across architectures and tasks while demonstrating robustness to label noise (Baek et al., 2024) and quantization (Na et al., 2022), making it an ideal choice for our merging settings where we don't have access to training data or labels. In MTL settings, as we demonstrate, favoring flat basins reduces cross-task sensitivity, allowing the merged model to generalize better.

## 3 METHODOLOGY

We develop `SAMerging` in three steps. First, we analyze the generalization of a merged model through a PAC-Bayes lens, which reveals a *cross-task heterogeneity* term that precisely captures expert-task mismatch. The bound suggests better generalization may be achieved in flatter basins of loss. Second, this analysis motivates casting coefficient learning as *multi-teacher knowledge distillation* on a small, unlabeled calibration set: as shown, minimizing merged-expert KL directly tightens an upper bound on the merged model's excess MTL risk. Third, we operationalize this objective with *Sharpness-Aware Minimization* (SAM), which seeks flatter basins that enable the merged model to generalize better across tasks. Together, these pieces yield a data-efficient, label-free procedure that avoids inference overhead while remaining robust to initialization. All proofs for the lemmas, propositions, and theorems are deferred to Appendix B.

### 3.1 GENERALIZATION OF THE MERGED MODEL

**Notation and Setup.** Let $[T] = \{1, \ldots, T\}$. Each task $t$ has data distribution $\mathcal{D}_t \subseteq \mathcal{Z}$ over $\mathcal{Z} = \mathcal{X} \times \mathcal{Y}$ with i.i.d. sample $\mathcal{S}_t = \{(x_i^{(t)}, y_i^{(t)})\}_{i=1}^{n_t} \sim \mathcal{D}_t$. Let $f_\theta : \mathcal{X} \to \widehat{\mathcal{Y}}$ be a model with

parameters $\theta \in \Theta$, and let $\ell : \widehat{\mathcal{Y}} \times \mathcal{Y} \to [0,1]$ be a $\gamma$-smooth loss convex in the model scores. The population and empirical risks on task $t$ are

$$\mathcal{L}_{\mathcal{D}_t}(\theta) = \mathbb{E}_{(x,y)\sim\mathcal{D}_t}\big[\ell(f_\theta(x),y)\big], \qquad \widehat{\mathcal{L}}_{\mathcal{S}_t}(\theta) = \frac{1}{n_t}\sum_{i=1}^{n_t} \ell\big(f_\theta(x_i^{(t)}), y_i^{(t)}\big).$$

*Merging objective.* Let $\boldsymbol{\alpha} \in \Delta^{T-1}$ weight the evaluation mixture across tasks and define,

$$\mathcal{L}_{\boldsymbol{\alpha}}(\theta) = \sum_{t=1}^{T} \alpha_t\, \mathcal{L}_{\mathcal{D}_t}(\theta), \qquad \widehat{\mathcal{L}}_{\boldsymbol{\alpha}}(\theta) = \sum_{t=1}^{T} \alpha_t\, \widehat{\mathcal{L}}_{\mathcal{S}_t}(\theta).$$

For a joint-trained MTL model $\theta_{\mathrm{MTL}}$, the objective is simply minimizing $\widehat{\mathcal{L}}_{\boldsymbol{\alpha}}(\theta_{\mathrm{MTL}})$. However, for merging, it is different as we have fine-tuned models. Given a pretrained $\theta_0$, fine-tuning on $\mathcal{S}_t$ yields $\theta_t$ and task vector $\tau_t = \theta_t - \theta_0$. Following task arithmetic (Ilharco et al., 2023), we form a merged model by layer-wise coefficients of the model $\lambda = \{\lambda_t^l\}_{t,l}$ and the objective of merging becomes:

$$\theta_\lambda^l = \theta_0^l + \sum_{t=1}^{T} \lambda_t^l\, \tau_t^l, \qquad \theta_{\mathrm{merge}} = \theta_{\lambda^\star}, \qquad \lambda^\star \in \arg\min_\lambda\, \widehat{\mathcal{L}}_{\boldsymbol{\alpha}}(\theta_\lambda).$$

**Deriving a multi-task PAC-Bayes generalization bound.** We analyze the generalization of the merged model via PAC-Bayes: here, $P$ is a data-free prior over model parameters and $Q$ is the posterior distribution after training on the data. As a common assumption in PAC-Bayes generalization, we assume that priors and posteriors are from multivariate Gaussian distributions (Dziugaite & Roy, 2017; Foret et al., 2021; Kim et al., 2025; Haddouche et al., 2025). Let $P := \mathcal{N}(\mu_p, \Sigma_P)$ be a data-free prior on $\Theta$ and let $Q_t := \mathcal{N}(\mu_t, \Sigma_t)$ be a posterior over $\Theta$ for task $t$. For posterior $Q$, define $\mathcal{L}_{\mathcal{D}_t}(Q) = \mathbb{E}_{\theta\sim Q}[\mathcal{L}_{\mathcal{D}_t}(\theta)]$ and $\widehat{\mathcal{L}}_{\mathcal{S}_t}(Q)$ analogously; if $Q_t$ are expert posteriors and $\boldsymbol{\beta} \in \Delta^{T-1}$, set $Q_{\mathrm{merge}} := \sum_{t=1}^{T} \beta_t Q_t$, and we aim to bound $\mathcal{L}_{\boldsymbol{\alpha}}(Q_{\mathrm{merge}})$. The layer-wise coefficients $\lambda$ are algorithmic (real-valued, possibly negative) and used only to construct $\theta_{\mathrm{merge}}$ via any merging method. In contrast, $\boldsymbol{\beta} \in \Delta^{T-1}$ are analytic simplex weights introduced for PAC-Bayes analysis over posteriors.

Our bounds will come with notions of flatness; to that end, we adopt the squared-gradient proxy to define flatness (Matena & Raffel, 2022; Haddouche et al., 2025). Specifically, for $\mathcal{S}_t = \{(x_i^{(t)}, y_i^{(t)})\}_{i=1}^{n_t} \sim \mathcal{D}_t$, define the population and empirical flatness as

$$\mathcal{G}_{\mathcal{D}_t}(\theta) = \mathbb{E}_{(x,y)\sim\mathcal{D}_t}\Big[\big\|\nabla_\theta \ell\big(f_\theta(x),y\big)\big\|_2^2\Big], \qquad \widehat{\mathcal{G}}_{\mathcal{S}_t}(\theta) = \frac{1}{n_t}\sum_{i=1}^{n_t}\big\|\nabla_\theta \ell\big(f_\theta(x_i^{(t)}), y_i^{(t)})\big)\big\|_2^2,$$

$$\mathcal{G}_{\mathcal{D}_t}(Q) = \mathbb{E}_{(x,y)\sim\mathcal{D}_t}\Big[\mathbb{E}_{h\sim Q}\big\|\nabla_h \ell\big(f_h(x),y\big)\big\|_2^2\Big], \widehat{\mathcal{G}}_{\mathcal{S}_t}(Q) = \frac{1}{n_t}\sum_{i=1}^{n_t}\mathbb{E}_{h\sim Q}\big\|\nabla_h \ell\big(f_h(x_i^{(t)}), y_i^{(t)})\big)\big\|_2^2.$$

**Lemma 1.** *For any task $t$ and posteriors $\{Q_j\}_{j=1}^{T}$, if $Q_{\mathrm{merge}} = \sum_j \beta_j Q_j$, then*

$$\mathcal{L}_{\mathcal{D}_t}(Q_{\mathrm{merge}}) = \sum_{j=1}^{T} \beta_j\, \mathcal{L}_{\mathcal{D}_t}(Q_j).$$

**Proposition 1.** *The multi-task risk $\mathcal{L}_{\boldsymbol{\alpha}}(Q_{\mathrm{merge}})$ can be decomposed as*

$$\mathcal{L}_{\boldsymbol{\alpha}}(Q_{\mathrm{merge}}) = \sum_{j=1}^{T} \beta_j\, \mathcal{L}_{\mathcal{D}_j}(Q_j) + \underbrace{\sum_{i=1}^{T}\sum_{j=1}^{T} \alpha_i \beta_j\, \big(\mathcal{L}_{\mathcal{D}_i}(Q_j) - \mathcal{L}_{\mathcal{D}_j}(Q_j)\big)}_{:=\mathcal{H}_Q(\boldsymbol{\alpha},\boldsymbol{\beta})}$$

The cross-task heterogeneity term $\mathcal{H}_Q(\boldsymbol{\alpha}, \boldsymbol{\beta})$ measures how much worse expert $Q_j$ performs on $\mathcal{D}_i$ compared to its own domain $\mathcal{D}_j$. It vanishes if $\mathcal{D}_i$ coincide as in the single-task setting or $Q_j \equiv Q$ for all $j$ as in a joint-trained MTL model. Now, we bound the $\sum_{j=1}^{T} \beta_j\, \mathcal{L}_{\mathcal{D}_j}(Q_j)$ using the following theorem.

**Theorem 1.** *Fix nonnegative $\{\delta_t\}_{t=1}^T$ such that $\delta = \sum_{t=1}^T \delta_t \le 1$. For any $\eta_t \in (0, 2)$ for each task $t$, any data-free prior $P = \mathcal{N}(\mu_P, \Sigma_P)$, any loss $\ell : \widehat{\mathcal{Y}} \times \mathcal{Y} \to [0, 1]$, with probability at least $1 - \delta$ over $\{S_t\}_{t=1}^T$ from $\{\mathcal{D}_t\}_{t=1}^T$ with $|S_t| = n_t$, for all $Q_t := \mathcal{N}(\theta_t, \Sigma_t)$,*

$$\mathcal{L}_{\boldsymbol{\alpha}}(Q_{\text{merge}}) \le \sum_{t=1}^T \beta_t \left[ \frac{1}{1 - \frac{\eta_t}{2}} \left( \hat{\mathcal{L}}_{S_t}(Q_t) + \frac{D_{\text{KL}}(Q_t \| P) + \log(\frac{1}{\delta_t})}{\eta_t n_t} \right) + \frac{\eta_t}{2 - \eta_t} \|\Sigma_t\| \, \mathcal{G}_{\mathcal{D}_t}(Q_t) \right]$$

$$+ \sum_{i=1}^T \sum_{j=1}^T \alpha_i \beta_j \left( \mathcal{L}_{\mathcal{D}_i}(Q_j) - \mathcal{L}_{\mathcal{D}_j}(Q_j) \right).$$

The bound on $\mathcal{L}_{\boldsymbol{\alpha}}(Q_{\text{merge}})$ decomposes into three parts: (i) *per-task PAC-Bayes terms* that require each expert $Q_t$ to generalize on its own domain, (ii) a *flatness penalty* $\mathcal{G}_{\mathcal{D}_t}(Q_t)$, which is small when the loss landscape is flat, and (iii) the *cross-task heterogeneity* which captures transfer mismatch across tasks. Hence, excess risk is controlled when experts are accurate in their own domains. Experts occupy flat basins with methods that encourage flatter minima and thus reduce $\mathcal{G}_{\mathcal{D}_t}(Q_t)$, and a small cross-task heterogeneity term.

**Going from posterior bound to a single model.** To pass from posterior-level guarantees to a single model in a non-convex landscape, we linearize the network at the pretrained point $\theta_0$ (NTK approximation as done in Jacot et al. (2020); Ortiz-Jimenez et al. (2023)). Let

$$\Phi(x) := \nabla_\theta f_{\theta_0}(x), \qquad f_\theta(x) = f_{\theta_0}(x) + \Phi(x)^\top (\theta - \theta_0),$$

for $\theta$ in a *neighborhood* of $\theta_0$. This makes the score affine in $\theta$ and induces the task kernels

$$\mathcal{K}_{\mathcal{D}_t} := \mathbb{E}_{(x,y) \sim \mathcal{D}_t}[\Phi(x)\Phi(x)^\top], \qquad \widehat{\mathcal{K}}_{S_t} := \frac{1}{n_t} \sum_{i=1}^{n_t} \Phi(x_i^{(t)})\Phi(x_i^{(t)})^\top.$$

Within this local model, convexity and $\gamma$-smoothness in the score let us relate Gaussian posteriors $Q_t = \mathcal{N}(\mu_t, \Sigma_t)$ to their means via trace terms in $\mathcal{K}$, enabling a single-model bound for the merged parameter

$$\theta_{\text{merge}} := \mathbb{E}\left[ Q_{\text{merge}} \right] = \mathbb{E}\left[ \sum_{j=1}^T \beta_j Q_j \right] = \sum_{j=1}^T \beta_j \, \mathbb{E}[Q_j] = \sum_{j=1}^T \beta_j \, \mu_j, \qquad \Delta_j := \mu_j - \theta_{\text{merge}}.$$

Note that posterior-level bounds themselves do not rely on NTK; the NTK is used only to pass from posterior-level to a single model. We now bound the loss and flatness of the posterior to later leverage in deriving the bound.

**Lemma 2.** *Assume the NTK linearization around $\theta_0$. Let $\ell$ be convex and $\gamma$-smooth in the score. For $Q = \mathcal{N}(\mu, \Sigma)$ and distribution $\mathcal{D}$,*

$$\mathcal{L}_{\mathcal{D}}(\mu) \le \mathcal{L}_{\mathcal{D}}(Q) \le \mathcal{L}_{\mathcal{D}}(\mu) + \frac{\gamma}{2} \operatorname{tr}(\Sigma \mathcal{K}_{\mathcal{D}}).$$

*The empirical statement follows verbatim with $\mathcal{D} \to \mathcal{S}$ and $\mathcal{K}_{\mathcal{D}} \to \widehat{\mathcal{K}}_{\mathcal{S}}$.*

**Lemma 3.** *Let $\ell$ be convex and $\gamma$-smooth in the score, and let $Q = \mathcal{N}(\mu, \Sigma)$. Then, for $\mathcal{D}$,*

$$\mathcal{G}_{\mathcal{D}}(Q) \le \left( \sqrt{\mathcal{G}_{\mathcal{D}}(\mu)} + \gamma \sqrt{\operatorname{tr}(\Sigma \mathcal{K}_{\mathcal{D}}^2)} \right)^2.$$

*The empirical statement follows verbatim with $\mathcal{D} \to \mathcal{S}$ and $\mathcal{K}_{\mathcal{D}} \to \widehat{\mathcal{K}}_{\mathcal{S}}$.*

Now, under NTK, we bound the heterogeneity term with loss and flatness at $\theta_{\text{merge}}$.

**Lemma 4.** *Under NTK, with loss $\ell$ being convex and $\gamma$-smooth in score, let $\mathcal{K}_\alpha = \sum_{t=1}^T \alpha_t \mathcal{K}_{\mathcal{D}_t}$, $\mathcal{K}_\beta = \sum_{j=1}^T \beta_j \mathcal{K}_{\mathcal{D}_j}$, and $\Delta_j = \mu_j - \theta_{\text{merge}}$. Then, for $\theta_{\text{merge}}$,*

$$\mathcal{H}_Q(\boldsymbol{\alpha}, \boldsymbol{\beta}) \le (\mathcal{L}_{\boldsymbol{\alpha}}(\theta_{\text{merge}}) - \mathcal{L}_{\boldsymbol{\beta}}(\theta_{\text{merge}}))$$

$$+ \sqrt{2 \left( \sum_{t=1}^T \alpha_t \mathcal{G}_{\mathcal{D}_t}(\theta_{\text{merge}}) + \sum_{j=1}^T \beta_j \mathcal{G}_{\mathcal{D}_j}(\theta_{\text{merge}}) \right)} \sqrt{\sum_{j=1}^T \beta_j \|\Delta_j\|_2^2}$$

$$+ \frac{\gamma}{2} \sum_{j=1}^T \beta_j \left[ \Delta_j^\top (\mathcal{K}_\alpha + \mathcal{K}_\beta) \Delta_j + \operatorname{tr}(\Sigma_j \mathcal{K}_{\boldsymbol{\alpha}}) \right].$$

With these, we are ready to state our main PAC-Bayes bound for the merged model.

**Theorem 2.** *Assume the NTK regime. Fix nonnegative $\{\delta_t\}_{t=1}^T$ with $\delta = \sum_{t=1}^T \delta_t \le 1$, task weights $\boldsymbol{\alpha}, \boldsymbol{\beta} \in \Delta^{T-1}$, constants $\eta_t \in (0,2)$ for each task $t$, a data-free prior $P \in \mathcal{M}(\mathcal{H})$, and a $\gamma$-smooth loss $\ell : \widehat{\mathcal{Y}} \times \mathcal{Y} \to [0,1]$ convex in its score argument. Over $\{S_t\}_{t=1}^T$ from $\{\mathcal{D}_t\}_{t=1}^T$ with $|S_t| = n_t$, with probability at least $1 - \delta$, the following holds for all Gaussian posteriors $Q_t = \mathcal{N}(\mu_t, \Sigma_t)$ and for the merged parameter $\theta_{\mathrm{merge}} := \sum_{j=1}^T \beta_j \mu_j$ with $\Delta_j := \mu_j - \theta_{\mathrm{merge}}$:*

$$\mathcal{L}_{\boldsymbol{\alpha}}(\theta_{\mathrm{merge}}) \le \sum_{t=1}^T \beta_t \left[ \frac{1}{1 - \frac{\eta_t}{2}} \left( \hat{\mathcal{L}}_{S_t}(\mu_t) + \frac{\gamma}{2} \operatorname{tr}(\Sigma_t \mathcal{K}_{\mathcal{D}_t}) + \frac{D_{\mathrm{KL}}(Q_t \| P) + \log\left(\frac{1}{\delta_t}\right)}{\eta_t n_t} \right) \right.$$

$$\left. + \frac{\eta_t}{2 - \eta_t} \|\Sigma_t\| \left( \sqrt{\mathcal{G}_{\mathcal{D}}(\mu_t)} + \gamma \sqrt{\operatorname{tr}(\Sigma_t \mathcal{K}_{\mathcal{D}_t}^2)} \right)^2 \right] + [\mathcal{L}_{\boldsymbol{\alpha}}(\theta_{\mathrm{merge}}) - \mathcal{L}_{\boldsymbol{\beta}}(\theta_{\mathrm{merge}})]$$

$$+ \sqrt{2 \left( \sum_{t=1}^T \alpha_t \mathcal{G}_{\mathcal{D}_t}(\theta_{\mathrm{merge}}) + \sum_{j=1}^T \beta_j \mathcal{G}_{\mathcal{D}_j}(\theta_{\mathrm{merge}}) \right)} \sqrt{\sum_{j=1}^T \beta_j \|\Delta_j\|_2^2}$$

$$+ \frac{\gamma}{2} \sum_{j=1}^T \beta_j \left[ \Delta_j^\top (\mathcal{K}_{\alpha} + \mathcal{K}_{\beta}) \Delta_j + \operatorname{tr}(\Sigma_j \mathcal{K}_{\boldsymbol{\alpha}}) \right]$$

*where $\mathcal{K}_{\boldsymbol{\alpha}} = \sum_{t=1}^T \alpha_t \mathcal{K}_{\mathcal{D}_t}$ and $\mathcal{K}_{\boldsymbol{\beta}} = \sum_{t=1}^T \beta_t \mathcal{K}_{\mathcal{D}_t}$.*

Theorem 2 implies that the risk $\mathcal{L}_{\boldsymbol{\alpha}}(\theta_{\mathrm{merge}})$ is controlled by two components: *(i) Per-task PAC-Bayes contributions* computed at each expert $Q_t = \mathcal{N}(\mu_t, \Sigma_t)$ that combine the expert's empirical loss with its flatness on its own domain via $\mathcal{G}_{\mathcal{D}_t}$. This component directly explains why using flatter experts improves merging, as in Lee et al. (2025), since flatter experts tighten these terms. *(ii) A cross-task heterogeneity contribution* that, under the NTK assumption, is further bounded by the mixture gap $\mathcal{L}_{\boldsymbol{\alpha}}(\theta_{\mathrm{merge}}) - \mathcal{L}_{\boldsymbol{\beta}}(\theta_{\mathrm{merge}})$, flatness of the merged model $\sum_t \alpha_t \mathcal{G}_{\mathcal{D}_t}(\theta_{\mathrm{merge}})$, the dispersion $\sum_j \beta_j \|\Delta_j\|_2^2$, and the quadratics $\sum_j \beta_j \Delta_j^\top (\mathcal{K}_{\boldsymbol{\alpha}} + \mathcal{K}_{\boldsymbol{\beta}}) \Delta_j$.

*Operationally*, the bound suggests desirable design choices for any label-free merging routine: (i) favoring flatter basins (e.g., via sharpness-aware perturbations) to directly shrink the flatness terms $\mathcal{G}_{\mathcal{D}_t}(\theta_{\mathrm{merge}})$, (ii) selecting coefficients $\boldsymbol{\beta}$ that pull $\theta_{\mathrm{merge}}$ to reduce dispersion and the kernel-weighted penalties, and (iii) aligning $\boldsymbol{\beta}$ with the evaluation mixture $\boldsymbol{\alpha}$ to minimize the gap $\mathcal{L}_{\boldsymbol{\alpha}}(\theta_{\mathrm{merge}}) - \mathcal{L}_{\boldsymbol{\beta}}(\theta_{\mathrm{merge}})$. Furthermore, we know that NTK approximation is best within a limited distance, so we encourage the merged model to be *around the pretrained* and not get too far. Taken together, these choices would tighten the bound. The bound also clarifies failure modes: experts that are simultaneously sharp and far from consensus increase the heterogeneity terms, and any principled algorithm should accordingly assign them smaller coefficients in $\boldsymbol{\beta}$.

### 3.2 THEORETICAL JUSTIFICATION FOR CASTING MODEL MERGING AS MULTI-TEACHER KNOWLEDGE DISTILLATION

We justify estimating the merging coefficients $\lambda$ via multi-teacher knowledge distillation: minimizing the KL divergence between the student's (merged model) and the teachers' (experts' models) distributions tightens a provable upper bound on the merged model's classification error. We first fix notation and losses, then recall standard links between distributional distances and 0–1 risk that underlie the single-task bound and its multi-task extension.

**Definition 1.** *Let $\mathcal{Y}$ be a finite label set and let $(x, y) \sim \mathcal{D}$. For each $x$, denote by $y(\cdot \mid x) \in \mathbb{R}^{|\mathcal{Y}|}$ the* true *conditional label distribution, by $p(\cdot \mid x)$ a (possibly misspecified) teacher/expert, and by $q(\cdot \mid x)$ a student/merged model. Let $h_q(x) \in \arg\max_{y \in \mathcal{Y}} q(y \mid x)$ be the deterministic classifier induced by $q$, and define the 0-1 risk under a conditional distribution $s(\cdot \mid x)$ by*

$$\mathcal{L}_s^{0-1}(h_q) := \mathbb{E}_{(x,y) \sim \mathcal{D}} \left[ 1 - s(h_q(x) \mid x) \right],$$

*and the Bayes optimal risk by*

$$\mathcal{L}_s^{0-1,\star} := \mathbb{E}_{(x,y) \sim \mathcal{D}} \left[ 1 - \max_y s(y \mid x) \right].$$

Now we propose the excess risk bound for a single task, which serves as the foundation for our main multi-task result.

**Lemma 5** (Single-Task Excess Risk Bound). *For any task $t$, let $y_t(\cdot|x)$ be the true data distribution, $p_t(\cdot|x)$ be a teacher, and $q_\lambda(\cdot|x)$ be the student. Let $h_\lambda$ be the classifier induced by the student. The student's excess* 0-1 *risk is bounded by:*

$$\mathcal{L}_{y_t}^{0-1}(h_\lambda) \leq \sqrt{2\,\mathbb{E}_{x\sim\mathcal{D}_t}\, D_{\mathrm{KL}}\big(p_t(\cdot|x)\,\|\,q_\lambda(\cdot|x)\big)} + \sqrt{2\,\mathbb{E}_{x\sim\mathcal{D}_t}\, D_{\mathrm{KL}}\big(y_t(\cdot|x)\,\|\,p_t(\cdot|x)\big)}.$$

Extending this result, we arrive at the main theorem in this section, which bounds the average excess risk of the merged model across all tasks.

**Theorem 3** (Multi-Task Excess Risk Bound). *Let there be $T$ tasks. For each task $t$, let $y_t(\cdot|x)$ be the true distribution, $p_t(\cdot|x)$ be the teacher (expert) for task $t$, and $q_\lambda(\cdot|x)$ be the student (merged model). Let $h_\lambda$ be the classifier induced by the student. For evaluation weights $\boldsymbol{\alpha} \in \Delta^{T-1}$, the student's average excess risk is bounded by:*

$$\sum_{t=1}^{T} \alpha_t \mathcal{L}_{y_t}^{0-1}(h_\lambda) \leq \sqrt{2\sum_{t=1}^{T}\alpha_t\mathbb{E}_{x\sim\mathcal{D}_t} D_{\mathrm{KL}}\big(p_t(\cdot|x)\,\|\,q_\lambda(\cdot|x)\big)} \tag{1}$$

$$+ \sqrt{2\sum_{t=1}^{T}\alpha_t\mathbb{E}_{x\sim\mathcal{D}_t} D_{\mathrm{KL}}\big(y_t(\cdot|x)\,\|\,p_t(\cdot|x)\big)}.$$

Theorem 3 decomposes the average excess risk into (i) an optimizable *student–teacher fit* term, given by the KL divergence $D_{\mathrm{KL}}(p_t\,\|\,q_\lambda)$, and (ii) a fixed *teacher error* term that depends only on the experts. Since the latter is independent of $\lambda$, tightening the bound reduces to minimizing the fit term. Our objective achieves this by minimizing the student–teacher KL divergence on calibration data, thereby directly tightening the proven risk bound for the merged model. In contrast, methods such as AdaMerging (Yang et al., 2023) use entropy minimization without an explicit excess-risk guarantee.

### 3.3 SAMERGING OBJECTIVE AND OPTIMIZATION

By Theorem 3, the average excess risk is controlled by the student–teacher fit term $D_{\mathrm{KL}}(p_t\|q_\lambda)$. Motivated by this and by the flatness terms in our PAC-Bayes bound in Theorem 2, we minimize the multi-teacher KD loss and search for *flat* solutions via SAM (Foret et al., 2021):

$$\mathcal{L}_{\mathrm{KD}}(\lambda) = \sum_{t=1}^{T} \alpha_t\, \mathbb{E}_{x\in\mathcal{B}_t}[\mathrm{KL}(p_t(\cdot\,|\,x)\,\|\,q_\lambda(\cdot\,|\,x))]$$

where $\mathcal{B}_t$ is a batch of unlabeled data for task $t$. The SAM-enhanced problem that we minimize is

$$\min_{\lambda}\ \max_{\|\varepsilon\|_2\leq\rho}\ \mathcal{L}_{\mathrm{KD}}(\lambda+\varepsilon),$$

where $\rho > 0$ controls the SAM neighborhood. In practice, we usually set $\alpha_t = \frac{1}{T}$. We can also initialize the coefficient from 0 or add a norm penalty to encourage the remaining in the NTK-faithful neighborhood of $\theta_0$, which helps reduce the dispersion and kernel-weighted penalties highlighted by Theorem 2. The pseudo-code of SAMerging is provided in Appendix C.

## 4 EXPERIMENTS

We first introduce our experimental setup in Section 4.1 and then report the results in Section 4.2. Full tasks and baseline descriptions (D.1) and tables (D.2) are provided in the appendix.

### 4.1 EXPERIMENTAL SETUP

**Tasks and data.** We evaluate generalization across increasing interference regimes on four suites following Ilharco et al. (2023); Wang et al. (2024): (i) **TA-8** (8 image classification tasks), (ii) **TALL-14** (TA-8 + six more tasks), (iii) **TALL-20** (TALL-14 + six more tasks), and (iv) **GLUE** (7

| Method | CLIP ViT-B/32 | | | | | | CLIP ViT-L/14 | | | | | |
| | TA-8 | | TALL-14 | | TALL-20 | | TA-8 | | TALL-14 | | TALL-20 | |
| | Acc. | Norm. | Acc. | Norm. | Acc. | Norm. | Acc. | Norm. | Acc. | Norm. | Acc. | Norm. |
|---|---|---|---|---|---|---|---|---|---|---|---|---|
| *Bases* | | | | | | | | | | | | |
| Pretrained | 48.2 | 53.4 | 56.9 | 64.3 | 55.6 | 61.9 | 64.6 | 68.5 | 69.1 | 74.0 | 65.6 | 70.2 |
| Fine-tuned | 90.3 | 100.0 | 88.5 | 100.0 | 89.8 | 100.0 | 94.3 | 100.0 | 93.4 | 100.0 | 93.5 | 100.0 |
| MTL | 88.5 | 98.0 | 87.7 | 99.1 | 88.9 | 99.0 | 92.3 | 98.0 | 91.6 | 98.1 | 91.8 | 98.2 |
| *Data-free* | | | | | | | | | | | | |
| Simple Averaging | 66.3 | 73.4 | 65.4 | 73.4 | 61.1 | 68.0 | 79.9 | 86.5 | 77.5 | 83.0 | 71.1 | 76.0 |
| Task Arithmetic | 67.5 | 74.8 | 66.5 | 75.1 | 61.1 | 68.0 | 82.1 | 87.1 | 77.9 | 83.4 | 71.1 | 76.0 |
| TIES-Merging | 71.9 | 79.6 | 67.6 | 76.4 | 62.7 | 69.8 | 83.8 | 88.9 | 77.8 | 83.3 | 72.3 | 77.3 |
| Isotropic Merging | 78.8 | 87.3 | 78.8 | 89.0 | 73.5 | 81.8 | 90.3 | 95.8 | 89.8 | 96.1 | 84.8 | 90.7 |
| PCB Merging | 75.4 | 83.5 | 70.3 | 79.4 | 64.1 | 71.4 | 84.2 | 89.3 | 80.4 | 86.1 | 72.6 | 77.6 |
| *Data-dependent* | | | | | | | | | | | | |
| Fisher ($k{=}1600$) | 70.5 | 78.1 | 67.1 | 75.8 | 62.2 | 69.3 | 73.3 | 77.7 | 75.4 | 80.7 | 70.4 | 75.3 |
| RegMean ($k{=}1600$) | 80.5 | 89.1 | 76.1 | 86.0 | 70.0 | 78.0 | 89.0 | 94.4 | 86.3 | 92.4 | 78.8 | 87.8 |
| RegMean++ ($k{=}1600$) | 84.2 | 93.2 | 79.8 | 90.2 | 74.0 | 82.4 | 88.3 | 93.6 | 87.9 | 94.1 | 82.5 | 88.2 |
| AdaMerging LW ($k{=}1600$) | 73.7 | 81.6 | 71.1 | 80.3 | 61.5 | 68.5 | 85.1 | 90.2 | 81.9 | 87.7 | 71.5 | 76.5 |
| AdaMerging LW ($k{=}16000$) | 82.6 | 91.5 | 77.7 | 87.8 | 69.4 | 77.3 | 91.0 | 96.5 | 87.2 | 93.4 | 79.0 | 84.5 |
| *Ours* | | | | | | | | | | | | |
| **SAMerging** ($k{=}1600$) | **87.1** | **96.5** | **83.7** | **94.6** | **81.1** | **90.3** | **92.6** | **98.2** | **90.7** | **97.1** | **89.9** | **96.1** |

Table 1: Average results across CLIP ViT backbones on TA-8, TALL-14, and TALL-20. Best result is **bold** and second best result is underlined. Accuracy (Acc., %) and normalized accuracy vs. Avg. fine-tuning (Norm., %; Avg. FT is 100%). Data-dependent methods use a maximum of $k$ unlabeled samples per task for adaptation.

NLP tasks). Vision backbones are CLIP ViT-B/32 and ViT-L/14. For NLP GLUE tasks, we use GPT-2, fine-tuned per task, to obtain task vectors. The setup is similar to Wang et al. (2024).

**Baselines and metric.** We compare against standard and state-of-the-art merging baselines. **Data-free**: Simple Averaging (SA) (Wortsman et al., 2022), Task Arithmetic (TA) (Ilharco et al., 2023), TIES-Merging (TIES) (Yadav et al., 2023), and Isotropic Merging (Marczak et al., 2025). **Data-dependent**: Fisher Merging (Matena & Raffel, 2022), RegMean (Jin et al., 2025) / RegMean++ (Nguyen et al., 2025), and AdaMerging (Yang et al., 2023) (we use the *layer-wise* variant, which consistently outperforms the task-wise version). We report average multi-task accuracy (Acc. %) and normalized accuracy relative to the mean accuracy of the individual experts (Norm. %).

### 4.2 RESULTS

**Overall performance.** Table 1 summarizes multi-task results across CLIP backbones. SAMerging achieves the best average accuracy and normalized accuracy across all settings, outperforming both data-free and data-dependent baselines. Notably, while AdaMerging (layer-wise) requires $k{=}16K$ unlabeled samples per task for adaptation, SAMerging uses only $k{=}1.6K$ and yields higher accuracy, indicating better data efficiency. Relative to strong pruning- or arithmetic-based methods, SAMerging consistently outperforms them and closes the gap to the single-task fine-tuned expert, while preserving the advantages of post-hoc merging (i.e., no joint training).

**Scaling with the number of tasks.** Table 1 shows that the absolute gains of SAMerging widen as the number of merged tasks increases. This trend supports our design objective: by explicitly stabilizing adaptation with sharpness-aware updates and a $D_{\mathrm{KL}}$ guidance term, SAMerging mitigates cross-task interference that typically accumulates with more experts.

**Language tasks.** As shown in Table 2, SAMerging achieves the highest average performance on GLUE with GPT-2 experts, surpassing TIES-Merging and Task Arithmetic; the same objective transfers beyond vision backbones to autoregressive language models and remains effective under heterogeneous task difficulty. Interestingly, data-dependent baselines (e.g., AdaMerging) underperform data-free ones (e.g., Task Arithmetic, TIES), underscoring the brittleness of entropy minimization.

**Ablation Study.** As shown in Table 4, we do an ablation study to find out the additive gain of performance by changing the optimizer and objective, and the KL objective and SAM each provide measurable gains, and their combination (KL+SAM) yields the strongest improvements. We also conduct ablation experiments to determine the effect of the number of examples on the data-

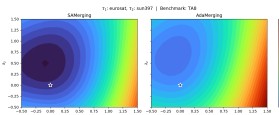
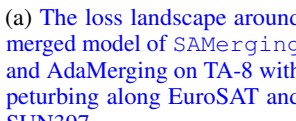
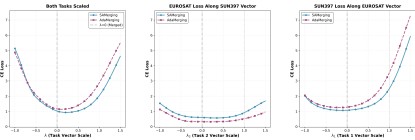
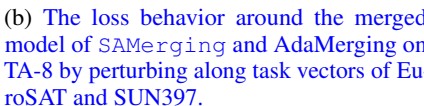
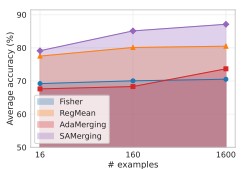

(a) The loss landscape around merged model of `SAMerging` and AdaMerging on TA-8 with peturbing along EuroSAT and SUN397.

(b) The loss behavior around the merged model of `SAMerging` and AdaMerging on TA-8 by perturbing along task vectors of EuroSAT and SUN397.

(c) Data-dependent methods gain in performance with increasing number of calibration data on TA-8 using ViT-B/32.

Figure 2: Performance analysis of model merging techniques under different conditions.

| Method | Avg. |
|---|---|
| Fine-tuned (STL) | 82.0 |
| Simple Average | 56.1 |
| Task Arithmetic ($\lambda$=0.5) | 70.0 |
| TIES-Merging ($\lambda$=0.6) | 70.0 |
| Fisher Merging | 58.7 |
| RegMean | 68.8 |
| AdaMerging | 68.8 |
| **SAMerging** | **74.9** |

Table 2: Merging methods performance on GLUE using GPT-2. Best result is **bold** and second best is underlined.

| Method | MNLI | IMDb | Avg. |
|---|---|---|---|
| Pre-trained | 33.0 | 50.0 | 41.5 |
| Finetuned | 91.7 | 97.1 | 94.4 |
| Task Arithmetic ($\lambda$=0.5) | 72.5 | 54.4 | 63.5 |
| Task Arithmetic ($\lambda$=1.0) | **91.2** | 65.9 | 78.6 |
| TIES-Merging | 88.7 | 95.9 | 92.3 |
| AdaMerging ($k$=1600) | 72.5 | 54.4 | 63.5 |
| SAMerging ($k$=1600) | 90.5 | **96.4** | **93.5** |

Table 3: Merging methods performance on MNLI and IMDb using DeBERTa-V2-XXL (1.5B parameters). Best result is **bold** and second best is underlined.

| Variant | Avg. Acc. (%) |
|---|---|
| `SAMerging` | **85.2** |
| – KL | 84.7 -0.5% ↓ |
| – SAM | 84.2 -1.0% ↓ |
| – KL & – SAM | 83.5 -1.7% ↓ |

Table 4: Ablation of using $D_{\mathrm{KL}}$ as objective and SAM as optimizer on TA-8 using ViT-B/32. Average Acc.% and drop vs. `SAMerging`.

dependent method, as shown in Figure 2c. The results show that `SAMerging` can achieve near SOTA performance with only a handful of data (e.g., 16 here).

## 5 CONCLUSION, LIMITATIONS, AND FUTURE WORK

We propose `SAMerging`, a data-efficient, label-free merger that learns layer-wise coefficients by explicitly seeking flat minima. We derive a PAC-Bayes bound for multi-task merging with a heterogeneity term that clarifies when merging succeeds, and cast coefficient learning as multi-teacher distillation, where minimizing student–teacher KL tightens the excess-risk bound of the merged model. Coupling this KL objective with SAM yields a procedure that generalizes well. Empirically, `SAMerging` achieves state-of-the-art results on TA-8 and TALL-14/20 with CLIP ViT and GLUE, consistently outperforming data-free and data-dependent baselines, without additional inference parameters or latency.

**Limitations and future work.** Even with more tasks, regimes with strong task interference or heavy domain shift (e.g., conflicting label spaces or multi-label settings) remain underexplored. The analysis assumes a local NTK-style linearization, so behavior far from this regime is uncertain. Our evaluation focuses solely on classification; extending it to generative tasks is left for future work. Finally, SAM adds calibration-time cost; lighter flatness proxies may reduce this overhead.

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

# A   THEORETICAL DEFINITION

**Definition 2** (Poincaré inequality, Malrieu & Talay (2006); Haddouche et al. (2025))**.** *A distribution $Q$ satisfies a Poincaré inequality with constant $c_P(Q)$ if for all sets of functions $f$ that are square-integrable, with their gradient's norm also being square-integrable, we have*

$$\mathrm{Var}_{h\sim Q}(f(h)) \leq c_P(Q)\mathbb{E}_{h\sim Q}[\|\nabla_h f(h)\|^2], \tag{2}$$

*where $\mathrm{Var}_{h\sim Q}(f(h)) = \mathbb{E}_{h\sim Q}[f(h) - \mathbb{E}_{h\sim Q}[f(h)]]^2$ is the variance of $f$ with respect to $Q$. We then say that $Q$ is Poincaré with constant $c_P(Q)$. A gaussian distribution $Q = \mathcal{N}(\mu, \Sigma)$ is Poincaré with constant $c_P(Q) = \|\Sigma\|$.*

**Definition 3** (Quadratically Self-Bounded, Haddouche et al. (2025))**.** *We say that $Q \in \mathcal{M}(\mathcal{H})$ is quadratically self-bounded with respect to the loss function $\ell \to \mathbb{R}_+$ and a constant $C > 0$ (namely $\mathrm{QSB}(\ell, C)$) if*

$$\mathbb{E}_{z\sim\mathcal{D}}\left[\left(\mathbb{E}_{h\sim Q}\,\ell(h,z)\right)^2\right] \;\leq\; C\mathcal{L}_\mathcal{D}(Q) \;=\; C\,\mathbb{E}_{z\sim\mathcal{D}}\left[\mathbb{E}_{h\sim Q}\,\ell(h,z)\right].$$

*Note that this is a relaxation of boundedness as if a loss is bounded $[0, C]$, the distribution $Q$ is $\mathrm{QSB}(\ell, C)$.*

# B   LEMMAS, THEOREMS, AND PROOFS

## B.1   PROOF OF LEMMA 1

*Proof.* We have $\mathcal{L}_{\mathcal{D}_t}(Q) = \int \mathcal{L}_{\mathcal{D}_t}(h)\,\mathrm{d}Q(h)$ and, by linearity of the integral, $\mathcal{L}_{\mathcal{D}_t}(Q_{\mathrm{merge}}) = \int \mathcal{L}_{\mathcal{D}_t}(h)\,\mathrm{d}\left(\sum_{j=1}^T \beta_j Q_j(h)\right) = \sum_{j=1}^T \beta_j \int \mathcal{L}_{\mathcal{D}_t}(h)\,\mathrm{d}Q_j(h) = \sum_{j=1}^T \beta_j \mathcal{L}_{\mathcal{D}_t}(Q_j).$ $\qquad\square$

## B.2   PROOF OF PROPOSITION 1

*Proof.* Using Lemma 1, we have

$$\mathcal{L}_{\mathrm{mix}}(Q_{\mathrm{merge}}) = \sum_{i=1}^T \alpha_i\,\mathcal{L}_{\mathcal{D}_i}(Q_{\mathrm{merge}}) = \sum_{i=1}^T\sum_{j=1}^T \alpha_i\beta_j\,\mathcal{L}_{\mathcal{D}_i}(Q_j).$$

Then by adding and subtracting $\sum_{j=1}^T \beta_j\,\mathcal{L}_{\mathcal{D}_j}(Q_j)$ and rearranging, we have

$$\mathcal{L}_{\mathrm{mix}}(Q_{\mathrm{merge}}) = \sum_{i=1}^T\sum_{j=1}^T \alpha_i\beta_j\,\mathcal{L}_{\mathcal{D}_i}(Q_j) + \left[\sum_{j=1}^T \beta_j\,\mathcal{L}_{\mathcal{D}_j}(Q_j) - \sum_{j=1}^T \beta_j\,\mathcal{L}_{\mathcal{D}_j}(Q_j)\right]$$

$$= \sum_{j=1}^T \beta_j\,\mathcal{L}_{\mathcal{D}_j}(Q_j) + \sum_{i=1}^T\sum_{j=1}^T \alpha_i\beta_j\left(\mathcal{L}_{\mathcal{D}_i}(Q_j) - \mathcal{L}_{\mathcal{D}_j}(Q_j)\right).$$

$$\square$$

**Theorem 4** (Theorem 6, Haddouche et al. (2025))**.** *Let $C > 0$, $\lambda \in \left(0, \frac{2}{C}\right)$, a prior $P$ over $\Theta$, a distribution $\mathcal{D}$, and a sample $\mathcal{S}_m \sim \mathcal{D}^m$. If a posterior $Q$ is Poincaeé with constant $c_P(Q)$ (Definition 2) and $\ell$ is $\mathrm{QSB}(\ell, C)$ under $Q$ (Definition 3), then with probability at least $1 - \delta$ over $\mathcal{S}_m$,*

$$\mathcal{L}_\mathcal{D}(Q) \;\leq\; \frac{1}{1 - \frac{\lambda C}{2}}\left(\widehat{\mathcal{L}}_{S_m}(Q) \;+\; \frac{D_{\mathrm{KL}}(Q\|P) + \log\left(\frac{1}{\delta}\right)}{\lambda m}\right) + \frac{\lambda}{2 - \lambda C}\,c_P(Q)\,\mathbb{E}_{z\sim D}\,\mathbb{E}_{h\sim Q}\,\|\nabla_h\ell(h,z)\|_2^2.$$

## B.3   PROOF OF THEOREM 1

*Proof.* We start from Proposition 1. We have:

$$\mathcal{L}_{\mathrm{mix}}(Q_{\mathrm{merge}}) = \sum_{j=1}^T \beta_j\,\mathcal{L}_{\mathcal{D}_j}(Q_j) + \sum_{i=1}^T\sum_{j=1}^T \alpha_i\beta_j\left(\mathcal{L}_{\mathcal{D}_i}(Q_j) - \mathcal{L}_{\mathcal{D}_j}(Q_j)\right).$$

We now bound $\sum_{j=1}^{T} \beta_j \mathcal{L}_{\mathcal{D}_j}(Q_j)$ term. For each $t \in [T]$, apply Theorem 4 with $m = n_t$, $\lambda = \eta_t$, $Q = Q_t = \mathcal{N}(\theta_t, \Sigma_t)$, and the same prior $P$; since $\ell \in [0, 1]$, QSB$(\ell, 1)$ holds *(i.e., $C = 1$)*, and for Gaussian $Q_t$, $c_P(Q_t) = \|\Sigma_t\|$. Hence, with probability at least $1 - \delta_t$,

$$\mathcal{L}_{D_t}(Q_t) \leq \frac{1}{1 - \frac{\eta_t}{2}} \left( \widehat{\mathcal{L}}_{S_t}(Q_t) + \frac{D_{\mathrm{KL}}(Q_t \| P) + \log\left(\frac{1}{\delta_t}\right)}{\eta_t n_t} \right) + \frac{\eta_t}{2 - \eta_t} \|\Sigma_t\| \, \mathcal{G}_{D_t}(Q_t),$$

Choose $\delta_t > 0$ such that $\sum_{t=1}^{T} \delta_t \leq \delta$. By a union bound over $t \in [T]$ and substituting the above inequality into the decomposition, with probability at least $1 - \delta$,

$$\mathcal{L}_\alpha(Q_{\mathrm{merge}}) \leq \sum_{t=1}^{T} \beta_t \left[ \frac{1}{1 - \frac{\eta_t}{2}} \left( \widehat{\mathcal{L}}_{S_t}(Q_t) + \frac{D_{\mathrm{KL}}(Q_t \| P) + \log\left(\frac{1}{\delta_t}\right)}{\eta_t n_t} \right) + \frac{\eta_t}{2 - \eta_t} \|\Sigma_t\|_{\mathrm{op}} \, \mathcal{G}_{D_t}(Q_t) \right]$$

$$+ \sum_{i=1}^{T} \sum_{j=1}^{T} \alpha_i \beta_j \left( \mathcal{L}_{\mathcal{D}_i}(Q_j) - \mathcal{L}_{\mathcal{D}_j}(Q_j) \right).$$

$\square$

## B.4 PROOF OF LEMMA 2

*Proof.* Given the convexity of $\ell$ and Jensen's inequality, we have $\mathcal{L}_{\mathcal{D}_t}(\mu_t) = \mathcal{L}_{\mathcal{D}_t}(\mathbb{E}_{h \sim Q_t}(h)) \leq \mathbb{E}_{h \sim Q_t}(\mathcal{L}_{\mathcal{D}_t}(h)) = \mathcal{L}_{\mathcal{D}_t}(Q_t)$. Fix $x, y$. Let $\Delta_h = f_h(x) - f_\mu(x) = \nabla_\theta f_{\theta_0}(x)^\top (h - \mu)$. With $\ell$ being convex, $\gamma$-smooth in scores

$$\ell(f_\mu(x) + \Delta_h, y) \leq \ell(f_\mu(x), y) + \langle \nabla_s \ell(f_\mu(x), y), \Delta_h \rangle + \frac{\gamma}{2} \|\Delta_h\|_2^2$$

Now if we take expectations over $(x, y) \sim \mathcal{D}$, the linear term $\langle \nabla_s \ell(f_\mu(x), y), \Delta_h \rangle$ will vanish. Now, plugging $\mathbb{E}_{x \sim \mathcal{D}} \mathbb{E}_{h \sim Q} \|\Delta_h\|_2^2 = \mathbb{E}_{x \sim \mathcal{D}} \mathbb{E}_{h \sim Q} \|\Phi(x)^\top (h - \mu)\|_2^2 = \mathrm{tr}(\Sigma \mathcal{K}_{\mathcal{D}})$ will give the bound. The proof for empirical follows the same procedure, but with the empirical observation of data. $\square$

## B.5 PROOF OF LEMMA 3

*Proof.* Based on the chain rule, we have,

$$\mathcal{G}_{\mathcal{D}}(Q) = \mathbb{E}_{(x, y) \sim \mathcal{D}} \left[ \mathbb{E}_{h \sim Q} \| \nabla_h \ell(f_h(x), y) \|_2^2 \right] = \mathbb{E}_{(x, y) \sim \mathcal{D}} \left[ \mathbb{E}_{h \sim Q} \| \Phi(x)^\top \nabla_s \ell(f_h(x), y) \|_2^2 \right].$$

We then add and subtract $\nabla_s \ell(f_\mu(x), y)$,

$$\mathcal{G}_{\mathcal{D}}(Q) = \mathbb{E}_{(x, y) \sim \mathcal{D}} \left[ \mathbb{E}_{h \sim Q} \| \Phi(x)^\top (\nabla_s \ell(f_\mu(x), y) + \nabla_s \ell(f_h(x), y) - \nabla_s \ell(f_\mu(x), y)) \|_2^2 \right].$$

By Minkowski's inequality,

$$\sqrt{\mathcal{G}_{\mathcal{D}}(Q)} = \left( \mathbb{E}_{(x, y) \sim \mathcal{D}} \mathbb{E}_{h \sim Q} \| \Phi(x)^\top (\nabla_s \ell(f_\mu(x), y) + \nabla_s \ell(f_h(x), y) - \nabla_s \ell(f_\mu(x), y)) \|_2^2 \right)^{\frac{1}{2}}$$

$$\leq \left( \mathbb{E}_{(x, y) \sim \mathcal{D}} \mathbb{E}_{h \sim Q} \| \Phi(x)^\top \nabla_s \ell(f_\mu(x), y) \|_2^2 \right)^{\frac{1}{2}}$$

$$+ \left( \mathbb{E}_{(x, y) \sim \mathcal{D}} \mathbb{E}_{h \sim Q} \| \Phi(x)^\top (\nabla_s \ell(f_h(x), y) - \nabla_s \ell(f_\mu(x), y)) \|_2^2 \right)^{\frac{1}{2}}$$

$$= \sqrt{\mathcal{G}_{\mathcal{D}}(\mu)} + \left( \mathbb{E}_{(x, y) \sim \mathcal{D}} \mathbb{E}_{h \sim Q} \| \Phi(x)^\top (\nabla_s \ell(f_h(x), y) - \nabla_s \ell(f_\mu(x), y)) \|_2^2 \right)^{\frac{1}{2}}.$$

Now, given the convexity and $\gamma$-smoothness of $\ell$ with respect to scores,

$$\| \nabla_s \ell(f_h(x), y) - \nabla_s \ell(f_\mu(x), y) \|_2 \leq \gamma |f_h(x) - f_\mu(x)| = \gamma |\Phi(x)^\top (h - \mu)|.$$

Hence,

$$\left( \mathbb{E}_{(x, y)} \mathbb{E}_h \| \Phi(x)^\top (\nabla_s \ell(f_h(x), y) - \nabla_s \ell(f_\mu(x), y)) \|_2^2 \right)^{\frac{1}{2}} \leq \gamma \left( \mathbb{E}_x \| \Phi(x) \|_2^2 \, \mathbb{E}_h \left[ (\Phi(x)^\top (h - \mu))^2 \right] \right)^{\frac{1}{2}}$$

$$= \gamma \left( \mathbb{E}_x \| \Phi(x) \|_2^2 \, \Phi(x)^\top \Sigma \, \Phi(x) \right)^{\frac{1}{2}} \leq \gamma \sqrt{\mathrm{tr}(\Sigma \mathcal{K}_{\mathcal{D}}^2)}.$$

Therefore,

$$\sqrt{\mathcal{G}_{\mathcal{D}}(Q)} \leq \sqrt{\mathcal{G}_{\mathcal{D}}(\mu)} + \gamma \sqrt{\mathrm{tr}(\Sigma \mathcal{K}_{\mathcal{D}}^2)},$$

and squaring both sides gives

$$\mathcal{G}_{\mathcal{D}}(Q) \leq \left( \sqrt{\mathcal{G}_{\mathcal{D}}(\mu)} + \gamma \sqrt{\mathrm{tr}(\Sigma \mathcal{K}_{\mathcal{D}}^2)} \right)^2.$$

The empirical version follows the same procedure with $\mathcal{D} \to \mathcal{S}$ and $\mathcal{K}_{\mathcal{D}} \to \widehat{\mathcal{K}}_{\mathcal{S}}$. $\qquad\square$

## B.6   LEMMA 6

Define the *deterministic* heterogeneity term

$$\mathcal{H}_{\mu}(\boldsymbol{\alpha}, \boldsymbol{\beta}) := \sum_{i=1}^{T}\sum_{j=1}^{T} \alpha_i \beta_j \left( \mathcal{L}_{\mathcal{D}_i}(\mu_j) - \mathcal{L}_{\mathcal{D}_j}(\mu_j) \right).$$

**Lemma 6.** *Under NTK, with loss $\ell$ being convex and $\gamma$-smooth in score, let $\mathcal{K}_{\alpha} = \sum_{t=1}^{T} \alpha_t \mathcal{K}_{\mathcal{D}_t}$,*

$$\mathcal{H}_Q(\boldsymbol{\alpha}, \boldsymbol{\beta}) \leq \mathcal{H}_{\mu}(\boldsymbol{\alpha}, \boldsymbol{\beta}) + \frac{\gamma}{2}\sum_{j=1}^{T} \beta_j \mathrm{tr}\left( \Sigma_j \mathcal{K}_{\alpha} \right).$$

*Proof.* Under NTK, with loss $\ell$ being convex and $\gamma$-smooth in score, given Lemma 2, we have:

$$\mathcal{H}_Q(\boldsymbol{\alpha}, \boldsymbol{\beta}) = \sum_{i=1}^{T}\sum_{j=1}^{T} \alpha_i \beta_j \left( \mathcal{L}_{\mathcal{D}_i}(Q_j) - \mathcal{L}_{\mathcal{D}_j}(Q_j) \right)$$

$$\leq \sum_{i=1}^{T}\sum_{j=1}^{T} \alpha_i \beta_j \left( \mathcal{L}_{\mathcal{D}_i}(\mu_j) + \frac{\gamma}{2}\mathrm{tr}\left( \Sigma_j \mathcal{K}_{\mathcal{D}_i} \right) - \mathcal{L}_{\mathcal{D}_j}(\mu_j) \right)$$

$$= \sum_{i=1}^{T}\sum_{j=1}^{T} \alpha_i \beta_j \left( \mathcal{L}_{\mathcal{D}_i}(\mu_j) - \mathcal{L}_{\mathcal{D}_j}(\mu_j) \right) + \frac{\gamma}{2}\sum_{j=1}^{T} \beta_j \mathrm{tr}\left( \Sigma_j \sum_{i=1}^{T} \alpha_i \mathcal{K}_{\mathcal{D}_i} \right)$$

$$= \mathcal{H}_{\mu}(\boldsymbol{\alpha}, \boldsymbol{\beta}) + \frac{\gamma}{2}\sum_{j=1}^{T} \beta_j \mathrm{tr}\left( \Sigma_j \mathcal{K}_{\alpha} \right)$$

$$\square$$

## B.7   PROOF OF LEMMA 4

*Proof.* We have:

$$\mathcal{H}_{\mu}(\boldsymbol{\alpha}, \boldsymbol{\beta}) = \sum_{i=1}^{T}\sum_{j=1}^{T} \alpha_i \beta_j \left( \mathcal{L}_{\mathcal{D}_i}(\mu_j) - \mathcal{L}_{\mathcal{D}_j}(\mu_j) \right)$$

$$= \sum_{j=1}^{T} \beta_j \left( \sum_{i=1}^{T} \alpha_i \left( \mathcal{L}_{\mathcal{D}_i}(\mu_j) \right) - \mathcal{L}_{\mathcal{D}_j}(\mu_j) \right)$$

$$= \sum_{j=1}^{T} \beta_j \left( \mathcal{L}_{\alpha}(\mu_j) - \mathcal{L}_{\mathcal{D}_j}(\mu_j) \right)$$

Fix $\theta_{\mathrm{merge}}$ and for a data distribution $\mathcal{D}$, define $g_{\mathcal{D}}(\theta_{\mathrm{merge}}) = \mathbb{E}_{(x,y)\sim\mathcal{D}}\nabla_\theta \ell(f_\theta(x), y)$, for every $\mu_j$, using descent lemma, we have:

$$\mathcal{L}_{\mathcal{D}}(\theta_{\mathrm{merge}}) + \langle g_{\mathcal{D}}(\theta_{\mathrm{merge}}), \Delta_j \rangle - \frac{\gamma}{2}\Delta_j^{\top}\mathcal{K}_{\mathcal{D}}\Delta_j \leq \mathcal{L}_{\mathcal{D}}(\mu_j) \leq \mathcal{L}_{\mathcal{D}}(\theta_{\mathrm{merge}}) + \langle g_{\mathcal{D}}(\theta_{\mathrm{merge}}), \Delta_j \rangle + \frac{\gamma}{2}\Delta_j^{\top}\mathcal{K}_{\mathcal{D}}\Delta_j$$

Next, we apply the first inequality to $\mathcal{L}_\mathcal{D} = \mathcal{L}_{\mathcal{D}_j}$ and the second inequality to $\mathcal{L}_\mathcal{D} = \mathcal{L}_\alpha$. We then have:

$$\mathcal{L}_\alpha(\mu_j) \leq \mathcal{L}_\alpha(\theta_{\text{merge}}) + \langle g_\alpha(\theta_{\text{merge}}), \Delta_j \rangle + \frac{\gamma}{2}\Delta_j^\top \mathcal{K}_\alpha \Delta_j,$$

$$\mathcal{L}_{\mathcal{D}_j}(\mu_j) \geq \mathcal{L}_{\mathcal{D}_j}(\theta_{\text{merge}}) + \langle g_j(\theta_{\text{merge}}), \Delta_j \rangle - \frac{\gamma}{2}\Delta_j^\top \mathcal{K}_{\mathcal{D}_j} \Delta_j.$$

Subtract and we have:

$$\mathcal{H}_\mu(\boldsymbol{\alpha}, \boldsymbol{\beta}) = \sum_{j=1}^T \beta_j \left( \mathcal{L}_\alpha(\mu_j) - \mathcal{L}_{\mathcal{D}_j}(\mu_j) \right)$$

$$\leq \sum_{j=1}^T \beta_j \left( \mathcal{L}_\alpha(\theta_{\text{merge}}) - \mathcal{L}_{\mathcal{D}_j}(\theta_{\text{merge}}) \right) + \sum_{j=1}^T \beta_j \langle g_\alpha(\theta_{\text{merge}}) - g_j(\theta_{\text{merge}}), \Delta_j \rangle +$$

$$\frac{\gamma}{2}\sum_{j=1}^T \beta_j \Delta_j^\top (\mathcal{K}_\alpha + \mathcal{K}_{\mathcal{D}_j})\Delta_j$$

We then bound the $\sum_{j=1}^T \beta_j \langle g_\alpha(\theta_{\text{merge}}) - g_j(\theta_{\text{merge}}), \Delta_j \rangle$ using Cauchy-Schwartz:

$$\sum_{j=1}^T \beta_j \langle g_\alpha(\theta_{\text{merge}}) - g_j(\theta_{\text{merge}}), \Delta_j \rangle \leq \sqrt{\sum_{j=1}^T \beta_j \|g_\alpha(\theta_{\text{merge}}) - g_j(\theta_{\text{merge}})\|_2^2} \sqrt{\sum_{j=1}^T \beta_j \|\Delta_j\|_2^2}$$

We now bound the first term:

$$\sum_{j=1}^T \beta_j \|g_\alpha(\theta_{\text{merge}}) - g_j(\theta_{\text{merge}})\|_2^2 = \sum_{j=1}^T \beta_j \left( \|g_\alpha(\theta_{\text{merge}})\|_2^2 - 2\langle g_\alpha(\theta_{\text{merge}}), g_j(\theta_{\text{merge}}) \rangle + \|g_j(\theta_{\text{merge}})\|_2^2 \right)$$

$$= \|g_\alpha(\theta_{\text{merge}})\|_2^2 - 2\langle g_\alpha(\theta_{\text{merge}}), g_\beta(\theta_{\text{merge}}) \rangle + \sum_{j=1}^T \beta_j \|g_j(\theta_{\text{merge}})\|_2^2$$

$$\leq 2\|g_\alpha(\theta_{\text{merge}})\|_2^2 + 2\sum_{j=1}^T \beta_j \|g_j(\theta_{\text{merge}})\|_2^2$$

$$\leq 2\sum_{i=1}^T \alpha_i \|g_i(\theta_{\text{merge}})\|_2^2 + 2\sum_{j=1}^T \beta_j \|g_j(\theta_{\text{merge}})\|_2^2.$$

Now, given Jensen's inequality, we have:

$$2\sum_{i=1}^T \alpha_i \|g_i(\theta_{\text{merge}})\|_2^2 + 2\sum_{j=1}^T \beta_j \|g_j(\theta_{\text{merge}})\|_2^2 \leq 2\left( \sum_{i=1}^T \alpha_i \mathcal{G}_{\mathcal{D}_i}(\theta_{\text{merge}}) + \sum_{j=1}^T \beta_j \mathcal{G}_{\mathcal{D}_j}(\theta_{\text{merge}}) \right)$$

So we have:

$$\mathcal{H}_\mu(\boldsymbol{\alpha}, \boldsymbol{\beta}) = \sum_{j=1}^T \beta_j \left( \mathcal{L}_\alpha(\mu_j) - \mathcal{L}_{\mathcal{D}_j}(\mu_j) \right)$$

$$\leq \sum_{j=1}^T \beta_j \left( \mathcal{L}_\alpha(\theta_{\text{merge}}) - \mathcal{L}_{\mathcal{D}_j}(\theta_{\text{merge}}) \right) + \sqrt{2\left( \sum_{i=1}^T \alpha_i \mathcal{G}_{\mathcal{D}_i}(\theta_{\text{merge}}) + \sum_{j=1}^T \beta_j \mathcal{G}_{\mathcal{D}_j}(\theta_{\text{merge}}) \right)} \sqrt{\sum_{j=1}^T \beta_j \|\Delta_j\|_2^2}$$

$$\frac{\gamma}{2}\sum_{j=1}^T \beta_j \Delta_j^\top (\mathcal{K}_\alpha + \mathcal{K}_{\mathcal{D}_j})\Delta_j.$$

Now, apply 6 to get the following,

$$
\mathcal{H}_Q(\boldsymbol{\alpha}, \boldsymbol{\beta}) \leq (\mathcal{L}_{\boldsymbol{\alpha}}(\theta_{\text{merge}}) - \mathcal{L}_{\boldsymbol{\beta}}(\theta_{\text{merge}})) + \sqrt{2\left(\sum_{t=1}^{T} \alpha_t \mathcal{G}_{\mathcal{D}_t}(\theta_{\text{merge}}) + \sum_{j=1}^{T} \beta_j \mathcal{G}_{\mathcal{D}_j}(\theta_{\text{merge}})\right)} \sqrt{\sum_{j=1}^{T} \beta_j \|\Delta_j\|_2^2}
$$

$$
+ \frac{\gamma}{2} \sum_{j=1}^{T} \beta_j \left[\Delta_j^{\top}(\mathcal{K}_{\alpha} + \mathcal{K}_{\beta})\Delta_j + \text{tr}\left(\Sigma_j \mathcal{K}_{\boldsymbol{\alpha}}\right)\right].
$$

which concludes the proof.

$\square$

### B.8  PROOF OF THEOREM 2

*Proof.* From Theorem 1, we have:

$$
\mathcal{L}_{\boldsymbol{\alpha}}(Q_{\text{merge}}) \leq \sum_{t=1}^{T} \beta_t \left[\frac{1}{1 - \frac{\eta_t}{2}}\left(\hat{\mathcal{L}}_{S_t}(Q_t) + \frac{D_{\text{KL}}(Q_t \| P) + \log(\frac{1}{\delta_t})}{\eta_t n_t}\right) + \frac{\eta_t}{2 - \eta_t}\|\Sigma_t\| \, \mathcal{G}_{\mathcal{D}_t}(Q_t)\right]
$$

$$
+ \sum_{i=1}^{T} \sum_{j=1}^{T} \alpha_i \beta_j \left(\mathcal{L}_{\mathcal{D}_i}(Q_j) - \mathcal{L}_{\mathcal{D}_j}(Q_j)\right).
$$

Then we apply Lemma 2 to $\hat{\mathcal{L}}_{\mathcal{S}_t}(Q_t)$ and Lemma 3 to $\mathcal{G}_{\mathcal{D}_t}(Q_t)$ for every $t \in$. Then we have:

$$
\mathcal{L}_{\boldsymbol{\alpha}}(Q_{\text{merge}}) \leq \sum_{t=1}^{T} \beta_t \left[\frac{1}{1 - \frac{\eta_t}{2}}\left(\hat{\mathcal{L}}_{S_t}(\mu_t) + \frac{\gamma}{2}\text{tr}\left(\Sigma_t \mathcal{K}_{\mathcal{D}_t}\right) + \frac{D_{\text{KL}}(Q_t \| P) + \log\left(\frac{1}{\delta_t}\right)}{\eta_t n_t}\right)\right.
$$

$$
\left. + \frac{\eta_t}{2 - \eta_t}\|\Sigma_t\| \left(\sqrt{\mathcal{G}_{\mathcal{D}}(\mu_t)} + \gamma \sqrt{\text{tr}\left(\Sigma_t \mathcal{K}_{\mathcal{D}_t}^2\right)}\right)^2\right] + \mathcal{H}_Q(\boldsymbol{\alpha}, \boldsymbol{\beta})
$$

Now, we apply Lemma 4. Then we have:

$$
\mathcal{L}_{\boldsymbol{\alpha}}(Q_{\text{merge}}) \leq \sum_{t=1}^{T} \beta_t \left[\frac{1}{1 - \frac{\eta_t}{2}}\left(\hat{\mathcal{L}}_{S_t}(\mu_t) + \frac{\gamma}{2}\text{tr}\left(\Sigma_t \mathcal{K}_{\mathcal{D}_t}\right) + \frac{D_{\text{KL}}(Q_t \| P) + \log\left(\frac{1}{\delta_t}\right)}{\eta_t n_t}\right)\right.
$$

$$
\left. + \frac{\eta_t}{2 - \eta_t}\|\Sigma_t\| \left(\sqrt{\mathcal{G}_{\mathcal{D}}(\mu_t)} + \gamma \sqrt{\text{tr}\left(\Sigma_t \mathcal{K}_{\mathcal{D}_t}^2\right)}\right)^2\right] + [\mathcal{L}_{\boldsymbol{\alpha}}(\theta_{\text{merge}}) - \mathcal{L}_{\boldsymbol{\beta}}(\theta_{\text{merge}})]
$$

$$
+ \sqrt{2\left(\sum_{t=1}^{T} \alpha_t \mathcal{G}_{\mathcal{D}_t}(\theta_{\text{merge}}) + \sum_{j=1}^{T} \beta_j \mathcal{G}_{\mathcal{D}_j}(\theta_{\text{merge}})\right)} \sqrt{\sum_{j=1}^{T} \beta_j \|\Delta_j\|_2^2}
$$

$$
+ \frac{\gamma}{2} \sum_{j=1}^{T} \beta_j \left[\Delta_j^{\top}(\mathcal{K}_{\alpha} + \mathcal{K}_{\beta})\Delta_j + \text{tr}\left(\Sigma_j \mathcal{K}_{\boldsymbol{\alpha}}\right)\right]
$$

In the end, we use Lemma 2's left inequality:

$$\mathcal{L}_{\boldsymbol{\alpha}}\left(\theta_{\mathrm{merge}}\right) = \sum_{t=1}^{T} \alpha_t \mathcal{L}_{\mathcal{D}_t}(\theta_{\mathrm{merge}}) \ \leq \ \sum_{t=1}^{T} \alpha_t \mathcal{L}_{\mathcal{D}_t}(\theta_{\mathrm{merge}}) = \mathcal{L}_{\boldsymbol{\alpha}}\left(Q_{\mathrm{merge}}\right) \ \leq$$

$$\sum_{t=1}^{T} \beta_t \left[ \frac{1}{1-\frac{\eta_t}{2}} \left( \hat{\mathcal{L}}_{S_t}\left(\mu_t\right) + \frac{\gamma}{2} \operatorname{tr}\left(\Sigma_t \mathcal{K}_{\mathcal{D}_t}\right) + \frac{D_{\mathrm{KL}}\left(Q_t \| P\right) + \log\left(\frac{1}{\delta_t}\right)}{\eta_t n_t} \right) \right.$$

$$\left. + \frac{\eta_t}{2-\eta_t} \|\Sigma_t\| \left( \sqrt{\mathcal{G}_{\mathcal{D}}(\mu_t)} + \gamma \sqrt{\operatorname{tr}\left(\Sigma_t \mathcal{K}_{\mathcal{D}_t}^2\right)} \right)^2 \right] + \left[\mathcal{L}_{\boldsymbol{\alpha}}(\theta_{\mathrm{merge}}) - \mathcal{L}_{\boldsymbol{\beta}}(\theta_{\mathrm{merge}})\right]$$

$$+ \sqrt{2\left(\sum_{t=1}^{T} \alpha_t \mathcal{G}_{\mathcal{D}_t}(\theta_{\mathrm{merge}}) + \sum_{j=1}^{T} \beta_j \mathcal{G}_{\mathcal{D}_j}(\theta_{\mathrm{merge}})\right)} \sqrt{\sum_{j=1}^{T} \beta_j \|\Delta_j\|_2^2}$$

$$+ \frac{\gamma}{2} \sum_{j=1}^{T} \beta_j \left[\Delta_j^\top (\mathcal{K}_\alpha + \mathcal{K}_\beta)\Delta_j + \operatorname{tr}\left(\Sigma_j \mathcal{K}_{\boldsymbol{\alpha}}\right)\right]$$

which concludes the proof. $\qquad\square$

### B.9 LEMMA 7

**Lemma 7.** *For a fixed input $x$, let $p(\cdot \mid x)$ and $q(\cdot \mid x)$ be two conditional probability distributions. Let $y^\star \in \arg\max_y p(y \mid x)$ and $\hat{y} \in \arg\max_y q(y \mid x)$ be the optimal predictions under these distributions. Then, $p(y^\star \mid x) - p(\hat{y} \mid x) \leq \|p(\cdot \mid x) - q(\cdot \mid x)\|_1$.*

*Proof.* We can write the difference as $p(y^\star) - p(\hat{y}) = (p(y^\star) - q(y^\star)) + (q(y^\star) - q(\hat{y})) + (q(\hat{y}) - p(\hat{y}))$. By definition, $q(\hat{y}) \geq q(y^\star)$, so the middle term is non-positive. The remaining two terms are bounded by their absolute values, the sum of which is at most $\sum_{y \in \mathcal{Y}} |p(y) - q(y)|$. $\qquad\square$

### B.10 LEMMA 8

**Lemma 8.** *For any two conditional distributions $s_1$ and $s_2$ over data distribution $\mathcal{D}$ and any classifier $h$, we have $|\mathcal{L}_{s_1}^{0-1}(h) - \mathcal{L}_{s_2}^{0-1}(h)| \leq \mathbb{E}_{x \sim \mathcal{D}}[\mathrm{TV}(s_1, s_2)]$ and $|\mathcal{L}_{s_1}^{0-1,\star} - \mathcal{L}_{s_2}^{0-1,\star}| \leq \mathbb{E}_{x \sim \mathcal{D}}[\mathrm{TV}(s_1, s_2)]$.*

*Proof.* For any fixed $x$ and label $y \in \mathcal{Y}$, we have $|s_1(y|x) - s_2(y|x)| \leq \mathrm{TV}(s_1, s_2)$. Taking $y = h(x)$ and averaging over $x$ yields the first inequality. The second follows because of Lemma 8. $\qquad\square$

### B.11 LEMMA 9

**Lemma 9** (Pinsker's Inequality, Csiszár & Körner (2015)). *For any two distributions $u, v$ on a finite set, their total variation is bounded by the KL divergence:* $\mathrm{TV}(u, v) \leq \sqrt{\frac{1}{2} D_{\mathrm{KL}}(u \| v)}$.

### B.12 PROOF OF LEMMA 5

*Proof.* For a single task, we drop the subscript $t$ for clarity. The student's excess risk under the true distribution $y$ can be decomposed by adding and subtracting terms related to the teacher distribution $p$: (Note that $\mathcal{L}_y^{0-1,\star} = 0$ by its definition)

$$\mathcal{L}_y^{0-1}(h_\lambda) - \mathcal{L}_y^{0-1,\star} = \left(\mathcal{L}_p^{0-1}(h_\lambda) - \mathcal{L}_p^{0-1,\star}\right) + \left[\left(\mathcal{L}_y^{0-1}(h_\lambda) - \mathcal{L}_p^{0-1}(h_\lambda)\right) + \left(\mathcal{L}_p^{0-1,\star} - \mathcal{L}_y^{0-1,\star}\right)\right].$$

The first component is the *student-teacher fit*, representing the student's excess risk relative to the teacher. The second component is the *teacher error*, which captures the error introduced by using $p$

as a proxy for $y$. We now bound each term.

$$
\begin{aligned}
\mathcal{L}_p^{0-1}(h_\lambda) - \mathcal{L}_p^{0-1,\star} &= \mathbb{E}_{x\sim\mathcal{D}}\left[\max_y p(y|x) - p(h_\lambda(x)|x)\right] \\
&\leq \mathbb{E}_{x\sim\mathcal{D}}\left[\|p(\cdot|x) - q_\lambda(\cdot|x)\|_1\right] \quad \text{(by Lemma 7)} \\
&\leq \mathbb{E}_{x\sim\mathcal{D}}\left[\sqrt{2D_{\mathrm{KL}}(p(\cdot|x)\|q_\lambda(\cdot|x))}\right] \quad \text{(by definition of TV and Lemma 9)} \\
&\leq \sqrt{2\mathbb{E}_{x\sim\mathcal{D}}\left[D_{\mathrm{KL}}(p(\cdot|x)\|q_\lambda(\cdot|x))\right]} \quad \text{(by Jensen's inequality)}
\end{aligned}
$$

$$
\begin{aligned}
\left(\mathcal{L}_y^{0-1}(h_\lambda) - \mathcal{L}_p^{0-1}(h_\lambda)\right) + \left(\mathcal{L}_p^{0-1,\star} - \mathcal{L}_r^{0-1,\star}\right) &\leq \left|\mathcal{L}_y^{0-1}(h_\lambda) - \mathcal{L}_p^{0-1}(h_\lambda)\right| + \left|\mathcal{L}_p^{0-1,\star} - \mathcal{L}_r^{0-1,\star}\right| \\
&\leq 2\mathbb{E}_{x\sim\mathcal{D}}[\mathrm{TV}(y(\cdot|x), p(\cdot|x))] \quad \text{(by Lemma 8)} \\
&\leq \mathbb{E}_{x\sim\mathcal{D}}\left[\sqrt{2D_{\mathrm{KL}}(y(\cdot|x)\|p(\cdot|x))}\right] \quad \text{(by Lemma 9)} \\
&\leq \sqrt{2\mathbb{E}_{x\sim\mathcal{D}}\left[D_{\mathrm{KL}}(y(\cdot|x)\|p(\cdot|x))\right]} \quad \text{(by Jensen's inequality)}
\end{aligned}
$$

Combining the two bounds yields the desired result. $\qquad\square$

### B.13 PROOF OF THEOREM 3

*Proof.* We begin with the single-task excess risk bound from Proposition 5, established for each task $t \in [T]$. We take the weighted average of this inequality across all tasks using the evaluation weights $\boldsymbol{\alpha} \in \Delta^{T-1}$: (Note that $\mathcal{L}_{y_t}^{0-1,\star} = 0$ by its definition)

$$
\begin{aligned}
\sum_{t=1}^T \alpha_t \mathcal{L}_{y_t}^{0-1}(h_\lambda) &\leq \sum_{t=1}^T \alpha_t \left(\sqrt{2\,\mathbb{E}_{x\sim\mathcal{D}_t}\, D_{\mathrm{KL}}\big(p_t(\cdot|x)\|q_\lambda(\cdot|x)\big)} + \sqrt{2\,\mathbb{E}_{x\sim\mathcal{D}_t}\, D_{\mathrm{KL}}\big(y_t(\cdot|x)\|p_t(\cdot|x)\big)}\right) \\
&= \sum_{t=1}^T \alpha_t \sqrt{2\,\mathbb{E}_{x\sim\mathcal{D}_t}\, D_{\mathrm{KL}}\big(p_t(\cdot|x)\|q_\lambda(\cdot|x)\big)} + \sum_{t=1}^T \alpha_t \sqrt{2\,\mathbb{E}_{x\sim\mathcal{D}_t}\, D_{\mathrm{KL}}\big(y_t(\cdot|x)\|p_t(\cdot|x)\big)}.
\end{aligned}
$$

Given Jensen's inequality, we have:

$$
\begin{aligned}
\sum_{t=1}^T \alpha_t \sqrt{2\,\mathbb{E}_{x\sim\mathcal{D}_t}\, D_{\mathrm{KL}}\big(p_t(\cdot|x)\|q_\lambda(\cdot|x)\big)} &\leq \sqrt{\sum_{t=1}^T \alpha_t \left(2\,\mathbb{E}_{x\sim\mathcal{D}_t}\, D_{\mathrm{KL}}\big(p_t(\cdot|x)\|q_\lambda(\cdot|x)\big)\right)} \\
&= \sqrt{2\sum_{t=1}^T \alpha_t \mathbb{E}_{x\sim\mathcal{D}_t} D_{\mathrm{KL}}\big(p_t(\cdot|x)\|q_\lambda(\cdot|x)\big)}
\end{aligned}
$$

and

$$
\begin{aligned}
\sum_{t=1}^T \alpha_t \sqrt{2\,\mathbb{E}_{x\sim\mathcal{D}_t}\, D_{\mathrm{KL}}\big(y_t(\cdot|x)\|p_t(\cdot|x)\big)} &\leq \sqrt{\sum_{t=1}^T \alpha_t \left(2\,\mathbb{E}_{x\sim\mathcal{D}_t}\, D_{\mathrm{KL}}\big(y_t(\cdot|x)\|p_t(\cdot|x)\big)\right)} \\
&= \sqrt{2\sum_{t=1}^T \alpha_t \mathbb{E}_{x\sim\mathcal{D}_t} D_{\mathrm{KL}}\big(y_t(\cdot|x)\|p_t(\cdot|x)\big)}.
\end{aligned}
$$

Combining these two inequalities gives the final multi-task bound as stated in the theorem. $\qquad\square$

## C PSEUDO-CODE OF SAMERGING

Algorithm 1 contains the pseudo-code of SAMerging.

**Algorithm 1** `SAMerging`: Sharpness-aware Model Merging

---

**Inputs:**
1: $\theta_0$: Pretrained model parameters.
2: $\{\theta_t\}_{t=1}^T$: Set of $T$ fine-tuned expert model parameters.
3: $\{\mathcal{B}_t\}_{t=1}^T$: Set of $T$ unlabeled calibration datasets for each task.
4: $\rho$: Neighborhood size for SAM.
5: $\eta$: Learning rate for optimizer.
6: $E$: Total number of epochs.
7: $\{\alpha_t\}_{t=1}^T$: Task loss weights (typically $\alpha_t = \frac{1}{T}$).

**Output:**
8: $\lambda = \{\lambda_t^l\}$: Layer-wise merging coefficients.

**Procedure:**

9: **function** CONSTRUCTMERGEDMODEL($\lambda, \theta_0, \{\tau_t\}_{t=1}^T$)
10:     Initialize $\theta_\lambda \leftarrow \theta_0$
11:     **for** each layer $l$ **do**
12:         $\theta_\lambda^l \leftarrow \theta_0^l + \sum_{t=1}^T \lambda_t^l \tau_t^l$
13:     **end for**
14:     **return** $\theta_\lambda$
15: **end function**

16: **function** KNOWLEDGEDISTILLATIONLOSS($\theta_{\mathrm{merge}}, \{\theta_t\}_{t=1}^T, \{\mathcal{B}_t\}_{t=1}^T, \{\alpha_t\}_{t=1}^T$)
17:     $\mathcal{L}_{\mathrm{total}} \leftarrow 0$
18:     **for** $t = 1$ to $T$ **do**
19:         $p_t(\cdot|x) \leftarrow \mathrm{Softmax}(f_{\theta_t}(x))$                             $\triangleright$ Teacher (expert) distribution
20:         $q_\lambda(\cdot|x) \leftarrow \mathrm{Softmax}(f_{\theta_{\mathrm{merge}}}(x))$                  $\triangleright$ Student (merged) distribution
21:         $\mathcal{L}_{\mathrm{task}} \leftarrow \mathbb{E}_{x \in \mathcal{B}_t} [D_{\mathrm{KL}}(p_t(\cdot|x) \,\|\, q_\lambda(\cdot|x))]$
22:         $\mathcal{L}_{\mathrm{total}} \leftarrow \mathcal{L}_{\mathrm{total}} + \alpha_t \cdot \mathcal{L}_{\mathrm{task}}$
23:     **end for**
24:     **return** $\mathcal{L}_{\mathrm{total}}$
25: **end function**

**Initialization:**
26: **for** $t = 1$ to $T$ **do**
27:     $\tau_t \leftarrow \theta_t - \theta_0$                                                   $\triangleright$ Calculate task vectors
28: **end for**
29: $\lambda \leftarrow \mathbf{0}$                                                   $\triangleright$ Initialize merging coefficients

**Optimization Loop:**
30: **for** epoch = 1 to $E$ **do**
31:     $\theta_\lambda \leftarrow \mathrm{ConstructMergedModel}(\lambda, \theta_0, \{\tau_t\})$
32:
33:                                          $\triangleright$ SAM Ascent Step: Find worst-case perturbation
34:     $\mathcal{L}_{KD}(\lambda) \leftarrow \mathrm{KnowledgeDistillationLoss}(\theta_\lambda, \{\theta_t\}, \{\mathcal{B}_t\}, \{\alpha_t\})$
35:     $g(\lambda) \leftarrow \nabla_\lambda \mathcal{L}_{KD}(\lambda)$
36:     $\epsilon \leftarrow \rho \frac{g(\lambda)}{\|g(\lambda)\|_2}$                                     $\triangleright$ Normalize gradient to find ascent direction
37:
38:                                          $\triangleright$ SAM Descent Step: Update on perturbed parameters
39:     $\theta_{\lambda+\epsilon} \leftarrow \mathrm{ConstructMergedModel}(\lambda + \epsilon, \theta_0, \{\tau_t\})$
40:     $\mathcal{L}_{KD}(\lambda + \epsilon) \leftarrow \mathrm{KnowledgeDistillationLoss}(\theta_{\lambda+\epsilon}, \{\theta_t\}, \{\mathcal{B}_t\}, \{\alpha_t\})$
41:     $g_{\mathrm{SAM}}(\lambda) \leftarrow \nabla_\lambda \mathcal{L}_{KD}(\lambda + \epsilon)$
42:
43:     $\lambda \leftarrow \lambda - \eta \cdot g_{\mathrm{SAM}}(\lambda)$                         $\triangleright$ Update coefficients (e.g., via SGD or Adam)
44: **end for**

45: **return** $\lambda^*$                                         $\triangleright$ Return the final optimized coefficients

# D EXPERIMENTS RESULTS AND ABLATION

## D.1 FULL DATA AND BASELINE SETUP

### D.1.1 TASKS AND DATA

We evaluate generalization across increasing interference regimes on four suites following Ilharco et al. (2023); Wang et al. (2024): (i) **TA-8** (8 image classification tasks: Cars (Krause et al., 2013), DTD (Cimpoi et al., 2014), EuroSAT (Helber et al., 2019), GTSRB (Stallkamp et al., 2011), MNIST (LeCun, 1998), RESISC45 (Cheng et al., 2017), SUN397 (Xiao et al., 2016), SVHN (Netzer et al., 2011)), (ii) **TALL-14** (Wang et al., 2024) (TA-8 + six: Oxford-102 Flowers (Nilsback & Zisserman, 2008), CIFAR-100 (Krizhevsky et al., 2009), PCAM (Veeling et al., 2018), STL-10 (Coates et al., 2011), Oxford-IIIT Pet (Parkhi et al., 2012), FER2013 (Goodfellow et al., 2013)), (iii) **TALL-20** (Wang et al., 2024) (TALL-14 + six: EMNIST (Cohen et al., 2017), CIFAR-10 (Krizhevsky et al., 2009), Food-101 (Bossard et al., 2014), Fashion-MNIST (Xiao et al., 2016), KMNIST (Clanuwat et al., 2018), RenderedSST2 (Socher et al., 2013)), and (iv) **GLUE** (Wang et al., 2019) (7 NLP tasks: CoLA (Warstadt et al., 2019), SST-2 (Socher et al., 2013), MRPC (Dolan & Brockett, 2005), QQP, MNLI (Williams et al., 2017), RTE, QNLI (Rajpurkar et al., 2016)). Vision backbones are CLIP ViT-B/32 and ViT-L/14; for GLUE, we use GPT-2 fine-tuned per task to obtain task vectors, mirroring the setup in Wang et al. (2024).

### D.1.2 BASELINES

We use the following baselines for comparison:

- Simple Averaging (Wortsman et al., 2022): this method averages the experts' parameters to achieve the merged model.
- Task Arithmetic (Ilharco et al., 2023): This method treats the difference of each expert from the fine-tuned model as a "task vector", then scales and adds these task vectors to make the merged model.
- TIES-Merging (Yadav et al., 2023): This method prunes small-magnitude updates, resolves sign conflicts across experts, and merges only weights that agree in sign to reduce interference.
- Isotropic Merging (Marczak et al., 2025): This method derives a shared subspace from the combined updates (SVD), makes it isotropic, then adds and orthogonalizes each expert's residual directions before the same isotropic scaling—reducing interference while preserving specialization.
- Fisher Merging (Matena & Raffel, 2022): parameter-wise weighted averaging where each weight is scaled by its Fisher information.
- PCB-Merging:
- RegMean (Jin et al., 2025): closed-form layerwise regression on unlabeled activations to match expert/ensemble logits; solve linear layers by least squares, average the remaining parameters.
- RegMean++ (Nguyen et al., 2025): RegMean's closed-form layerwise regression, but compute Gram stats from activations of the partially merged model (not each expert), capturing cross-layer dependencies.
- AdaMerging (Yang et al., 2023): adaptively learns task-/layer-wise merge coefficients on unlabeled data by minimizing prediction entropy.

### D.1.3 EXPERIMENTS SETUP

Here we explain the setup in detail for each baseline and also `SAMerging`. Note that all experiments are conducted through the Fusion Bench benchmarking Tang et al. (2024):

- `SAMerging`: For training `SAMerging`, we set the learning rate to $0.001$; we use the SAM optimizer with Adam as its base optimizer with momentum $0.99$ and weight decay $5 \times 10^{-4}$; the perturbation radius is $\rho = 0.07$; batch size is 16; and weights are tied.

| Method | 0.05 | 0.07 | 0.1 | 0.15 | 0.2 | 0.3 |
|---|---|---|---|---|---|---|
| Task Arithmetic (TALL-14) | 64.0 | 65.3 | **66.5** | 66.2 | 63.3 | 52.8 |
| TIES-Merging (TALL-14) | 62.4 | 63.9 | 65.7 | 67.8 | **68.7** | 67.6 |
| Task Arithmetic (TALL-20) | 61.1 | **61.3** | 60.6 | 56.1 | 49.6 | 36.3 |
| TIES-Merging (TALL-20) | 60.6 | 61.6 | 62.7 | **63.0** | 61.8 | 55.5 |

Table 5: Average accuracy on ViT-B/32 for TALL-14 (14 tasks) and TALL-20 (20 tasks) model pools across different scaling factors $\lambda$.

| Method | 0.1 | 0.2 | 0.3 | 0.4 | 0.5 | 0.6 | 0.7 | 0.8 | 0.9 | 1.0 |
|---|---|---|---|---|---|---|---|---|---|---|
| Task Arithmetic | 64.1 | **69.5** | 67.5 | 60.7 | 51.3 | 42.4 | 35.1 | 29.3 | 24.4 | 19.3 |
| TIES-Merging | 60.6 | 67.6 | 71.9 | **73.1** | 71.7 | 68.4 | 64.0 | 59.0 | 53.5 | 48.6 |

Table 6: Average accuracy across TA-8 tasks on ViT-B/32 for different $\lambda$ values. Bold values indicate the best $\lambda$ for each method.

- MTL: We fine-tune CLIP with learning rate $1 \times 10^{-5}$, weight decay 0, seed 42; batch size 128 for TALL-14 over 4,000 steps and batch size 64 for TALL-20 over 8,000 steps (LoRA disabled).
- Simple Averaging: This method does not have any hyperparameters. It is an unweighted average of task models.
- Task Arithmetic:
  - TA-8: we set the scaling factor to 0.2 according to Table 6.
  - TALL-14: we set the scaling factor to 0.1 according to Table 5.
  - TALL-20: we set the scaling factor to 0.05 according to Table 5.
- TIES-Merging:
  - TA-8: we set the scaling factor to 0.4 according to Table 6 and top-k threshold to 20.
  - TALL-14: we set the scaling factor to 0.15 according to Table 5 and top-k threshold to 20..
  - TALL-20: we set the scaling factor to 0.15 according to Table 5 and top-k threshold to 20.
- Isotropic Merging: We use ISO-CTS variant with a scaling factor 1.0; we set the common-space fraction is 0.8.
- PBC-Merging: We set the parameter competition balancing ratio to 0.05.
  - TA-8: We set the scaling factor to 1.2 (default value).
  - TA-8: We set the scaling factor to 0.6.
  - TA-8: We set the scaling factor to 0.5.
- Fisher (k=1600): We compute Fisher weights with $k = 1600$ examples, normalize Fisher weights, set minimal Fisher weight to $1 \times 10^{-6}$, and use dataloader batch size 16 with 4 workers.
- RegMean: We set $k = 1600$ examples and set the reduce_off_diagonal to 0.6 (default value).
- RegMean++: We set $k = 1600$ examples and set the reduce_off_diagonal to 0.95 (default value)
- AdaMerging: We train AdaMerging with learning rate 0.001 using Adam and batch size 16; weights are tied and initialized to 0.2–0.3 depending on the setting.

## D.2 FULL EXPERIMENTS RESULTS

Below are the full results of merging methods' performance on each task for different suites and backbones.

| Method | 0.02 | 0.05 | 0.1 | 0.2 | 1.0 |
|---|---|---|---|---|---|
| Accuracy (%) | 81.77 | 81.82 | 81.82 | 81.82 | 81.80 |
| CE Loss | 0.649 | 0.648 | 0.648 | 0.647 | 0.648 |

Table 7: Average accuracy and cross-entropy (CE) loss on ViT-B/32 for TA8 model pool across different $\rho$ values.

| Backbone | Method | SUN397 | Cars | RESISC45 | EuroSAT | SVHN | GTSRB | MNIST | DTD | Avg. |
|---|---|---|---|---|---|---|---|---|---|---|
| ViT-B/32 | *Base* | | | | | | | | | |
| | MTL | 72.2 | 76.5 | 92.0 | 97.2 | 95.5 | 97.7 | 99.3 | 77.5 | 88.5 |
| | *Data-free* | | | | | | | | | |
| | Simple Averaging | 65.4 | 62.4 | 70.6 | 75.7 | 64.5 | 55.0 | 86.3 | 50.6 | 66.3 |
| | Task Arithmetic | 57.0 | 55.7 | 64.7 | 73.3 | 77.9 | 68.5 | 96.1 | 47.1 | 67.5 |
| | TIES-Merging | 67.0 | 64.2 | 74.3 | 74.5 | 77.7 | 69.4 | 94.1 | 54.0 | 71.9 |
| | Isotropic Merging | 71.6 | 73.6 | 84.1 | 87.1 | 73.0 | 80.9 | 95.3 | 65.0 | 78.8 |
| | *Data-dependent* | | | | | | | | | |
| | Fisher (k=1600) | 67.5 | 68.0 | 70.2 | 75.4 | 81.9 | 54.9 | 90.3 | 56.0 | 70.5 |
| | RegMean (k=1600) | 67.9 | 68.6 | 82.5 | 94.4 | 90.0 | 78.8 | 97.7 | 64.0 | 80.5 |
| | RegMean++ | 69.2 | 69.7 | 87.1 | 95.8 | 94.4 | 89.8 | 99.0 | 68.6 | 84.2 |
| | AdaMerging LW (k=1600) | 61.5 | 61.2 | 71.3 | 86.9 | 83.9 | 76.7 | 97.4 | 50.5 | 73.7 |
| | AdaMerging LW (k=16000) | 68.0 | 71.3 | 83.7 | 92.0 | 87.5 | 93.3 | 98.2 | 67.2 | 82.6 |
| | *Ours* | | | | | | | | | |
| | SAMerging (k=1600) | 71.1 | 75.0 | 91.3 | 96.6 | 92.4 | 96.8 | 98.1 | 75.6 | 87.1 |
| ViT-L/14 | *Base* | | | | | | | | | |
| | MTL | 79.0 | 89.3 | 94.4 | 98.3 | 96.4 | 98.1 | 99.4 | 83.7 | 92.3 |
| | *Data-free* | | | | | | | | | |
| | Simple Averaging | 72.5 | 81.5 | 82.3 | 88.5 | 81.6 | 74.0 | 96.6 | 61.8 | 79.9 |
| | Task Arithmetic | 73.3 | 81.4 | 84.1 | 89.6 | 86.6 | 81.7 | 97.6 | 62.3 | 82.1 |
| | TIES-Merging | 74.8 | 83.2 | 86.5 | 89.7 | 89.7 | 85.2 | 97.8 | 63.9 | 83.8 |
| | Isotropic Merging | 79.5 | 91.0 | 93.9 | 96.3 | 91.4 | 94.5 | 98.6 | 77.1 | 90.3 |
| | *Data-dependent* | | | | | | | | | |
| | Fisher (k=1600) | 70.0 | 79.2 | 70.3 | 99.0 | 65.0 | 58.8 | 85.5 | 58.4 | 73.3 |
| | RegMean (k=1600) | 75.4 | 88.2 | 91.0 | 96.7 | 95.8 | 92.6 | 98.5 | 73.6 | 89.0 |
| | RegMean++ | 77.5 | 89.6 | 68.2 | 97.3 | 97.0 | 96.3 | 99.1 | 81.4 | 88.3 |
| | AdaMerging LW (k=1600) | 74.5 | 83.5 | 86.6 | 92.4 | 90.9 | 90.7 | 98.2 | 63.7 | 85.1 |
| | AdaMerging LW (k=16000) | 78.1 | 90.7 | 90.7 | 96.1 | 95.0 | 97.6 | 98.6 | 81.3 | 91.0 |
| | *Ours* | | | | | | | | | |
| | SAMerging (k=1600) | 80.5 | 92.1 | 95.3 | 97.4 | 95.5 | 98.1 | 99.1 | 82.7 | 92.6 |

Table 8: TA-8 per-task accuracies (Acc., %). Columns list the 8 TA-8 tasks and Avg. Acc. is the mean over them.

We report per-task accuracies (Acc., %) for TA-8, TALL-14, and TALL-20 using CLIP ViT-B/32 and CLIP ViT-L/14. "Avg.' denotes the mean over tasks. For data-dependent methods, $k$ indicates the number of unlabeled samples per task used for adaptation. See Table 8, Table 9, and Table 10. We also report the results for the GLUE benchmark, as shown in Table 11

# E    USE OF LARGE LANGUAGE MODELS

We have utilized Large Language Models to assist us in writing, identifying related works, and initial ideation of model merging problems.

# F    VISUALIZATION

| Backbone | Method | SUN397 | Cars | RESISC45 | EuroSAT | SVHN | GTSRB | MNIST | DTD | Flowers | PCAM | FER2013 | Pet | STL10 | CIFAR100 | Avg. |
|---|---|---|---|---|---|---|---|---|---|---|---|---|---|---|---|---|
| ViT-B/32 | *Base* | | | | | | | | | | | | | | | |
| | MTL | 73.0 | 72.9 | 93.2 | 98.5 | 96.4 | 97.7 | 99.6 | 76.7 | 87.4 | 86.1 | 71.4 | 90.9 | 97.8 | 86.4 | 87.7 |
| | *Data-free* | | | | | | | | | | | | | | | |
| | Simple Averaging | 64.8 | 60.4 | 67.1 | 67.0 | 50.7 | 45.6 | 76.6 | 46.9 | 67.4 | 65.2 | 51.6 | 84.2 | 97.2 | 70.4 | 65.4 |
| | Task Arithmetic | 64.4 | 59.6 | 67.3 | 67.8 | 54.0 | 50.0 | 80.7 | 48.0 | 66.1 | 69.8 | 53.1 | 84.2 | 96.6 | 69.2 | 66.5 |
| | TIES-Merging | 62.2 | 54.6 | 65.3 | 63.0 | 65.7 | 64.3 | 92.6 | 49.9 | 58.2 | 77.1 | 54.9 | 81.4 | 94.8 | 62.4 | 67.6 |
| | Isotropic Merging | 70.6 | 68.8 | 81.6 | 85.1 | 73.5 | 81.2 | 96.4 | 61.9 | 75.3 | 80.7 | 66.5 | 88.9 | 97.5 | 75.3 | 78.8 |
| | *Data-dependent* | | | | | | | | | | | | | | | |
| | Fisher (k=1600) | 65.7 | 64.3 | 66.9 | 65.1 | 61.4 | 46.3 | 79.1 | 49.9 | 70.1 | 63.1 | 52.0 | 87.3 | 97.2 | 71.0 | 67.1 |
| | RegMean (k=1600) | 66.3 | 64.6 | 76.6 | 90.3 | 78.3 | 65.7 | 94.8 | 56.9 | 71.8 | 81.8 | 61.7 | 87.9 | 96.9 | 71.9 | 76.1 |
| | RegMean++ | 67.3 | 66.6 | 81.9 | 94.4 | 91.3 | 80.3 | 98.1 | 61.9 | 74.9 | 76.0 | 64.2 | 90.1 | 97.5 | 73.4 | 79.8 |
| | AdaMerging LW (k=1600) | 62.7 | 58.5 | 69.0 | 82.2 | 73.6 | 62.5 | 95.5 | 50.3 | 62.8 | 75.3 | 58.3 | 83.6 | 95.2 | 65.5 | 71.1 |
| | AdaMerging LW (k=16000) | 66.5 | 69.4 | 82.4 | 92.6 | 85.6 | 89.9 | 97.8 | 61.0 | 73.8 | 51.4 | 64.7 | 87.5 | 96.7 | 68.6 | 77.7 |
| | *Ours* | | | | | | | | | | | | | | | |
| | SAMerging (k=1600) | 68.9 | 72.2 | 90.6 | 93.8 | 89.1 | 94.0 | 98.5 | 72.4 | 83.4 | 77.6 | 67.6 | 90.2 | 97.2 | 76.6 | 83.7 |
| ViT-L/14 | *Base* | | | | | | | | | | | | | | | |
| | MTL | 79.2 | 88.9 | 94.8 | 98.0 | 96.1 | 97.5 | 99.3 | 83.8 | 97.5 | 90.8 | 72.9 | 96.1 | 99.4 | 88.4 | 91.6 |
| | *Data-free* | | | | | | | | | | | | | | | |
| | Simple Averaging | 71.2 | 79.0 | 78.7 | 80.4 | 71.3 | 64.6 | 94.3 | 58.7 | 81.9 | 74.2 | 54.8 | 94.6 | 99.3 | 82.4 | 77.5 |
| | Task Arithmetic | 71.6 | 78.4 | 79.3 | 80.3 | 72.4 | 67.9 | 95.3 | 59.8 | 81.9 | 78.2 | 54.8 | 94.8 | 99.0 | 82.3 | 77.9 |
| | TIES-Merging | 72.0 | 75.6 | 76.5 | 69.7 | 77.2 | 75.1 | 96.7 | 57.8 | 79.6 | 78.2 | 59.9 | 94.7 | 98.4 | 77.7 | 77.8 |
| | Isotropic Merging | 79.1 | 90.5 | 94.2 | 95.8 | 91.1 | 94.6 | 98.8 | 76.2 | 96.9 | 84.4 | 71.8 | 96.6 | 99.6 | 88.2 | 89.8 |
| | *Data-dependent* | | | | | | | | | | | | | | | |
| | Fisher (k=1600) | 70.0 | 77.4 | 75.9 | 97.1 | 60.4 | 57.7 | 86.6 | 58.1 | 84.2 | 59.8 | 52.8 | 94.7 | 99.4 | 81.0 | 75.4 |
| | RegMean (k=1600) | 72.9 | 84.7 | 87.0 | 95.2 | 92.9 | 86.3 | 98.1 | 66.9 | 92.0 | 86.6 | 66.1 | 96.1 | 99.3 | 83.5 | 86.3 |
| | RegMean++ | 74.1 | 86.7 | 89.7 | 96.7 | 95.7 | 91.7 | 98.9 | 71.1 | 94.2 | 80.2 | 70.0 | 96.1 | 99.3 | 85.7 | 87.9 |
| | AdaMerging LW (k=1600) | 73.7 | 80.5 | 84.6 | 88.7 | 84.6 | 83.3 | 97.5 | 62.4 | 83.9 | 69.2 | 61.7 | 95.4 | 98.9 | 81.4 | 81.9 |
| | AdaMerging LW (k=16000) | 77.5 | 90.0 | 91.2 | 96.1 | 94.3 | 96.2 | 98.5 | 77.0 | 95.3 | 51.3 | 74.0 | 95.9 | 99.4 | 83.3 | 87.2 |
| | *Ours* | | | | | | | | | | | | | | | |
| | SAMerging (k=1600) | 78.5 | 89.5 | 94.4 | 97.6 | 94.5 | 97.2 | 98.9 | 81.6 | 97.2 | 86.3 | 72.2 | 95.6 | 99.1 | 86.5 | 90.7 |

Table 9: TALL-14 per-task accuracies (Acc., %). Columns list the 14 TALL-14 tasks and Avg. Acc. is the mean over them.

| Backbone | Method | SUN397 | Cars | RESISC45 | EuroSAT | SVHN | GTSRB | MNIST | DTD | Flowers | PCAM | FER2013 | Pet | STL10 | CIFAR100 | CIFAR10 | Food101 | Fashion | EMNIST | KMNIST | SST2 | Avg. |
|---|---|---|---|---|---|---|---|---|---|---|---|---|---|---|---|---|---|---|---|---|---|---|
| ViT-B/32 | *Base* | | | | | | | | | | | | | | | | | | | | | |
| | MTL | 73.5 | 74.9 | 93.8 | 99.0 | 96.6 | 97.6 | 99.5 | 78.1 | 87.9 | 87.4 | 71.6 | 90.9 | 98.0 | 87.2 | 97.4 | 86.4 | 94.3 | 95.4 | 97.6 | 71.7 | 88.9 |
| | *Data-free* | | | | | | | | | | | | | | | | | | | | | |
| | Simple Averaging | 64.2 | 59.6 | 64.8 | 60.9 | 47.3 | 43.1 | 71.8 | 46.4 | 66.5 | 63.9 | 50.2 | 84.1 | 97.0 | 69.8 | 92.7 | 79.7 | 71.3 | 15.0 | 11.4 | 61.8 | 61.1 |
| | Task Arithmetic | 64.2 | 59.6 | 64.8 | 60.9 | 47.3 | 43.1 | 71.8 | 46.4 | 66.5 | 63.9 | 50.2 | 84.1 | 97.0 | 69.8 | 92.7 | 79.7 | 71.3 | 15.0 | 11.4 | 61.8 | 61.1 |
| | TIES-Merging | 65.0 | 59.7 | 64.6 | 60.7 | 52.4 | 49.1 | 79.3 | 48.4 | 66.7 | 66.7 | 51.7 | 84.1 | 97.0 | 70.4 | 93.3 | 79.5 | 72.4 | 17.3 | 12.2 | 61.9 | 62.7 |
| | Isotropic Merging | 68.0 | 59.2 | 76.9 | 81.8 | 73.9 | 80.8 | 96.5 | 58.5 | 72.7 | 83.5 | 64.6 | 86.3 | 96.9 | 73.9 | 95.0 | 75.9 | 83.2 | 35.3 | 37.2 | 69.8 | 73.5 |
| | *Data-dependent* | | | | | | | | | | | | | | | | | | | | | |
| | Fisher (k=1600) | 65.0 | 62.8 | 64.4 | 59.1 | 53.8 | 43.3 | 71.5 | 48.3 | 68.6 | 62.1 | 50.1 | 86.2 | 97.1 | 70.4 | 93.7 | 80.4 | 70.8 | 16.8 | 13.1 | 67.4 | 62.2 |
| | RegMean (k=1600) | 65.5 | 62.4 | 74.5 | 85.4 | 70.4 | 60.0 | 89.3 | 54.3 | 70.1 | 81.4 | 60.1 | 86.2 | 94.1 | 80.9 | 94.1 | 80.9 | 77.9 | 20.3 | 31.8 | 67.5 | 70.0 |
| | RegMean++ | 66.2 | 64.5 | 79.3 | 92.6 | 87.3 | 73.3 | 93.8 | 57.7 | 72.4 | 73.6 | 63.4 | 88.4 | 97.1 | 72.5 | 94.9 | 83.1 | 82.7 | 28.7 | 40.4 | 67.1 | 74.0 |
| | AdaMerging LW (k=1600) | 58.9 | 47.6 | 61.7 | 72.4 | 62.6 | 54.0 | 94.3 | 44.8 | 55.9 | 70.5 | 54.4 | 79.4 | 93.7 | 62.5 | 90.6 | 64.4 | 75.5 | 18.8 | 13.5 | 54.5 | 61.5 |
| | AdaMerging LW (k=16000) | 66.6 | 67.4 | 81.9 | 91.7 | 80.7 | 87.9 | 92.7 | 60.3 | 73.3 | 52.2 | 64.9 | 85.8 | 96.9 | 69.6 | 91.2 | 78.3 | 70.8 | 15.8 | 10.0 | 50.0 | 69.4 |
| | *Ours* | | | | | | | | | | | | | | | | | | | | | |
| | SAMerging (k=1600) | 66.1 | 67.8 | 86.6 | 94.7 | 81.8 | 90.1 | 95.2 | 68.6 | 78.9 | 72.2 | 63.7 | 89.2 | 96.0 | 71.2 | 92.0 | 78.8 | 87.8 | 81.2 | 88.8 | 71.1 | 81.1 |
| ViT-L/14 | *Base* | | | | | | | | | | | | | | | | | | | | | |
| | MTL | 79.2 | 89.2 | 95.1 | 98.2 | 96.1 | 97.7 | 99.3 | 82.9 | 98.0 | 90.7 | 72.7 | 95.7 | 99.5 | 87.9 | 98.5 | 92.3 | 92.1 | 93.0 | 91.2 | 85.7 | 91.8 |
| | *Data-free* | | | | | | | | | | | | | | | | | | | | | |
| | Simple Averaging | 70.7 | 77.7 | 76.4 | 75.3 | 69.5 | 62.1 | 93.7 | 57.7 | 80.8 | 73.6 | 52.7 | 94.2 | 99.2 | 81.7 | 97.0 | 90.5 | 77.4 | 16.1 | 10.4 | 66.1 | 71.1 |
| | Task Arithmetic | 70.7 | 77.7 | 76.4 | 75.3 | 69.5 | 62.1 | 93.7 | 57.7 | 80.8 | 73.6 | 52.7 | 94.2 | 99.2 | 81.7 | 97.0 | 90.5 | 77.4 | 16.1 | 10.4 | 66.1 | 71.1 |
| | TIES-Merging | 71.7 | 77.9 | 78.1 | 75.8 | 73.8 | 66.6 | 95.4 | 59.1 | 81.4 | 72.4 | 55.2 | 94.7 | 99.1 | 82.1 | 97.4 | 90.5 | 80.8 | 18.4 | 10.8 | 65.2 | 72.3 |
| | Isotropic Merging | 78.7 | 87.6 | 93.7 | 94.6 | 90.4 | 94.2 | 98.7 | 75.1 | 97.0 | 85.4 | 70.9 | 96.5 | 99.5 | 87.7 | 98.5 | 92.4 | 90.7 | 46.2 | 40.8 | 78.5 | 84.8 |
| | *Data-dependent* | | | | | | | | | | | | | | | | | | | | | |
| | Fisher (k=1600) | 70.0 | 77.2 | 76.1 | 96.9 | 61.3 | 57.7 | 87.5 | 58.1 | 83.8 | 60.1 | 52.1 | 94.6 | 99.4 | 81.0 | 96.9 | 89.9 | 74.7 | 15.4 | 10.1 | 64.7 | 70.4 |
| | RegMean (k=1600) | 71.7 | 82.6 | 84.6 | 94.1 | 89.0 | 80.0 | 97.1 | 64.1 | 90.0 | 84.6 | 63.1 | 95.6 | 99.3 | 82.5 | 97.6 | 91.5 | 86.9 | 32.9 | 19.9 | 68.3 | 78.8 |
| | RegMean++ | 73.1 | 85.3 | 88.2 | 96.3 | 94.5 | 88.2 | 98.0 | 68.2 | 92.1 | 80.8 | 68.4 | 96.2 | 99.2 | 84.3 | 98.2 | 91.8 | 88.9 | 45.4 | 42.8 | 70.3 | 82.5 |
| | AdaMerging LW (k=1600) | 71.8 | 73.3 | 76.1 | 71.1 | 76.0 | 72.6 | 97.4 | 57.4 | 78.9 | 64.8 | 59.3 | 94.5 | 98.1 | 76.7 | 96.2 | 83.4 | 81.5 | 18.3 | 11.7 | 66.4 | 71.5 |
| | AdaMerging LW (k=16000) | 76.7 | 89.5 | 89.3 | 96.1 | 91.9 | 95.4 | 98.2 | 74.3 | 94.2 | 51.6 | 69.8 | 95.6 | 99.2 | 82.7 | 96.4 | 90.1 | 86.0 | 12.8 | 10.0 | 79.7 | 79.0 |
| | *Ours* | | | | | | | | | | | | | | | | | | | | | |
| | SAMerging (k=1600) | 77.5 | 88.6 | 93.9 | 96.9 | 92.8 | 96.4 | 97.5 | 79.9 | 96.7 | 84.8 | 71.8 | 95.7 | 99.0 | 84.5 | 97.0 | 91.1 | 90.2 | 87.0 | 92.9 | 83.6 | 89.9 |

Table 10: TALL-20 per-task accuracies (Acc., %). Columns list the 20 TALL-20 tasks and Avg. Acc. is the mean over them.

| Backbone | Method | CoLA | MNLI | MRPC | QNLI | QQP | RTE | SST-2 | Avg. |
|---|---|---|---|---|---|---|---|---|---|
| GPT-2 | *Reference Results* | | | | | | | | |
| | Fine-tuned (STL) | 76.8 | 82.1 | 80.4 | 88.3 | 89.6 | 65.3 | 91.2 | 82.0 |
| | *Model Merging* | | | | | | | | |
| | Simple Average | 55.0 | 55.1 | 51.0 | 57.6 | 76.7 | 44.8 | 52.5 | 56.1 |
| | Task Arithmetic (λ=0.5) | 68.7 | 68.6 | 69.6 | 70.5 | 81.8 | 47.3 | 83.6 | 70.0 |
| | TIES-Merging (λ=0.6) | 68.4 | 71.4 | 68.4 | 69.6 | 82.4 | 47.7 | 81.8 | 70.0 |
| | Fisher Merging | 54.8 | 58.0 | 39.5 | 63.3 | 81.5 | 49.1 | 64.7 | 58.7 |
| | RegMean | 61.7 | 70.4 | 65.4 | 69.7 | 78.8 | 56.0 | 79.7 | 68.8 |
| | AdaMerging | 67.8 | 59.2 | 70.6 | 63.4 | 80.6 | 47.3 | 74.0 | 68.8 |
| | *Ours* | | | | | | | | |
| | **SAMerging** | 68.4 | 75.1 | 73.5 | 80.7 | 78.3 | 57.8 | 86.9 | 74.9 |

Table 11: Multi-task model merging methods using GPT-2 models on the GLUE benchmark.

| Method | CLIP ViT-B/32 | | | | | | CLIP ViT-L/14 | | | | | |
|---|---|---|---|---|---|---|---|---|---|---|---|---|
| | TA-8 | | TALL-14 | | TALL-20 | | TA-8 | | TALL-14 | | TALL-20 | |
| | Acc. | Norm. | Acc. | Norm. | Acc. | Norm. | Acc. | Norm. | Acc. | Norm. | Acc. | Norm. |
| Fine-tuned | 90.3 | 100.0 | 88.5 | 100.0 | 89.8 | 100.0 | 94.3 | 100.0 | 93.4 | 100.0 | 93.5 | 100.0 |
| MTL | 88.5 | 98.0 | 87.7 | 99.1 | 88.9 | 99.0 | 92.3 | 97.9 | 91.6 | 98.1 | 91.8 | 98.2 |
| ProDistill ($k{=}16$) | 81.1 | 89.8 | 80.5 | 91.0 | 77.8 | 86.6 | 87.2 | 92.5 | 89.0 | 95.3 | **86.8** | **92.8** |
| **SAMerging** ($k{=}16$) | **83.8** | **92.8** | **81.2** | **91.8** | **77.9** | **85.7** | **91.2** | **96.7** | **89.1** | **95.4** | 85.8 | 91.8 |

Table 12: Comparison of `SAMerging` ProDistill on CLIP ViT-B/32 and CLIP ViT-L/14

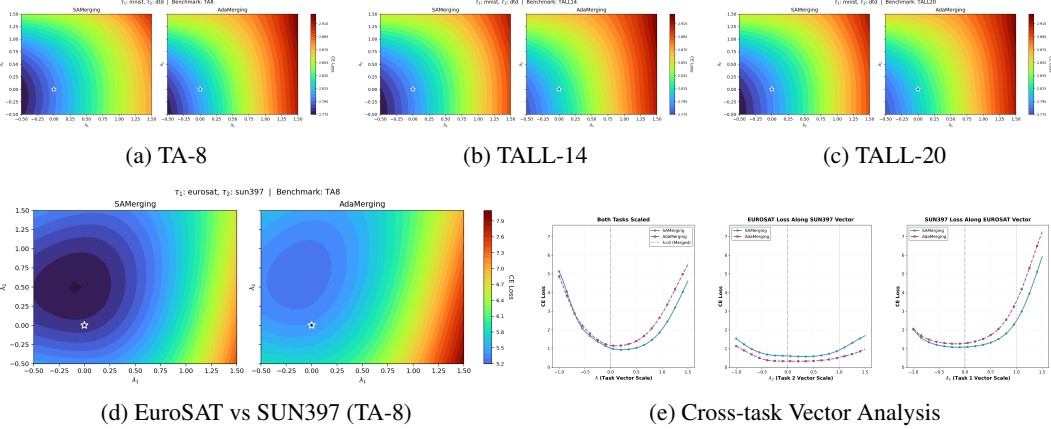

(a) TA-8         (b) TALL-14         (c) TALL-20

(d) EuroSAT vs SUN397 (TA-8)     (e) Cross-task Vector Analysis

Figure 3: **Loss Landscape Comparisons. (Top)** Loss surfaces for MNIST and DTD across TA-8, TALL-14, and TALL-20 benchmarks. **(Bottom)** Detailed analysis of EuroSAT and SUN397 on TA-8, showing the loss landscape (left) and loss behavior along specific task vectors (right).

