# OpenReview forum: "SAMerging: Sharpness-aware Model Merging via Multi-Teacher Knowledge Distillation"
_ICLR.cc/2026/Conference — Submitted to ICLR 2026_

### Official Review · Reviewer_fHM9 · 2025-10-21

**Soundness:** 3
**Presentation:** 3
**Contribution:** 2
**Rating:** 4
**Confidence:** 3

**Summary:**

First, the authors derive a PAC-Bayes generalization bound for the merged model, which highlights the importance of finding "flat" minima in the loss landscape and introduces a "cross-task heterogeneity" term to quantify the mismatch between different models. Second, they frame the model merging problem as a multi-teacher knowledge distillation task. This involves minimizing the KL divergence between the merged model's predictions and the expert models' predictions on a small, unlabeled dataset. To find solutions that generalize well, they employ Sharpness-Aware Minimization, which explicitly seeks out flat regions in the loss landscape.

**Strengths:**

* The derivation of the PAC-Bayes bound (Theorem 2) and the excess risk bound (Theorem 3) logically connects the goals of finding flat minima and minimizing KL divergence to the ultimate objective of better generalization. The results convincingly demonstrate the superiority of SAMerging over existing data-dependent methods, particularly in its data efficiency.
* The connection between the theoretical insights and the final algorithm is explained logically, making the design choices easy to follow.
* By providing a robust, data-efficient method, SAMerging addresses key limitations of prior work, such as sensitivity to initialization and high data requirements. It achieves state-of-the-art performance with minimal calibration data and no inference overhead.

**Weaknesses:**

* The theoretical proofs and methods used in this paper, such as SAM and multi-teacher knowledge distillation, have been widely explored in previous work. The contribution here appears to be more of a combination of existing techniques applied to model merging, rather than a novel approach.
* The paper acknowledges this limitation, noting that the NTK regime is most accurate near the pretrained initialization. It remains unclear how well this assumption holds in practice, especially when the fine-tuned models have diverged significantly from the pretrained model.
* The experiments are primarily focused on classification tasks. While these are standard benchmarks, the paper's claims would be strengthened by evaluating SAMerging on a more diverse set of tasks, particularly on modern LLMs.

**Questions:**

It adds computational cost during the merging process due to the use of SAM, which involves an additional forward/backward pass to find the "worst-case" perturbation. The authors acknowledge this as a "calibration-time cost" but do not quantify it relative to other methods. A brief analysis of this trade-off would provide a more complete picture of the method's practicality.

---

> ### Author Response · Authors · 2025-11-22
> **Response to Reviewer**
>
> We sincerely thank the reviewer for their thoughtful assessment. We address the noted weaknesses and questions below and will gladly incorporate the suggested improvements into the revised version to enhance the clarity of our work.
>
> **Q1: Contribution**
>
> Thanks for pointing this. We would like to point out our contribution in two aspects:
>    - To our knowledge, we are the first to derive a PAC-Bayes generalization bound that captures the complexity of the model-merging setting, beyond [1], which analyzes a simplified generalization and does not explicitly account for cross-task heterogeneity and the dependence on fine-tuned experts. In merging, we mix fine-tuned models, not data, which makes the analysis fundamentally different from standard single-model or joint multi-task generalization bounds [2, 3]. Because we do not have access to the underlying training data, we cannot rely on a standard empirical risk minimization framework on the merged model; instead, we must implicitly account for data heterogeneity *given* the expert models, which leads to the cross-task heterogeneity term in our bound.
>    - The second aspect involved moving from empirical observations to establishing theoretical guarantees when selecting the loss. By that, we take a step further compared to AdaMerging on why we should minimize KL for a theoretically-grounded bound of excess risk. That being said, this was not merely transitioning from entropy to KL but rather we *derived why we should minimize KL loss from scratch* which is interesting by its own.
>
> **Q2: NTK**
> Thanks for pointing this out. We did the following experiments to find out how far do merged models and expert models can have from the pre-trained model. We calculate the frobenius norm of the difference between the merged/expert models and the pre-trained model, and found that the merged model and the expert models are relatively very close to the pre-trained model. Results are in the following table:
>
> | Model | Dataset | Pretrained Norm | FT (avg.) | FT (rel.) | SAMerging | SAM (rel.) | AdaMerging | Ada (rel.) |
> |-------|---------|----------------|----------|----------|-----------|-----------|------------|----------|
> | ViT-B/32 | TA8 | 335.81 | 2.33 | 0.007 | 1.92 | 0.006 | 3.20 | 0.010 |
> | ViT-B/32 | TALL14 |  335.81 | 2.23 | 0.007 | 2.02 | 0.006 | 3.31 | 0.010 |
> | ViT-B/32 | TALL20 | 335.81 | 2.37 | 0.007 | 2.75 | 0.008 | 3.72 | 0.011 |
>
> | Model | Dataset | Pretrained Norm | FT (avg.) | FT (rel.) | SAMerging | SAM (rel.) | AdaMerging | Ada (rel.) |
> |-------|---------|----------------|----------|----------|-----------|-----------|------------|----------|
> | ViT-L/14 | TA8 | 492.24 | 3.90 | 0.008 | 3.57 | 0.007 | 5.22 | 0.011 |
> | ViT-L/14 | TALL14 | 492.24 | 3.88 | 0.008 | 4.17 | 0.008 | 5.49 | 0.011 |
> | ViT-L/14 | TALL20 | 492.24 | 4.07 | 0.008 | 5.60 | 0.011 | 6.32 | 0.013 |
>
> You can see that the merged models and the expert models are relatively very close to the pre-trained model. Furthermore, as we move to the large model ViT-L/14, the relative distance decreases. Note that the initialization of SAMerging is the pretrained model. This allows for earlier steps to be almost identical to the pre-trained model, respecting the NTK assumption. As you can see, the merged model of SAMerging is closer to the pretrained model than the AdaMerging model, in some cases even closer than the expert models, while AdaMerging is constantly further away.
>
> **Q3: Other Baselines**
> We are conducting experiments on LLMs and will report the results ASAP.
>
> **Q1 in Questions: Time Overhead**
> Here is the time overhead for each merging method for ViT-B/32 backbone on TA-8: (Results are in (mm:ss) format, omit trivially simpler methods like Simple Averaging, Task Arithmetic, and TIES). However, note that if we want SAMerging to have the same *performance* as others, we need a shorter training time.
> | Method        | TA-8 | TALL-14 | TALL-20 |
>    |---------------|-------------------|----------------------|----------------------|
>    | Fisher Merging    | 9:19           | 13:52                | 18:01                |
>    | RegMean           | 3:03           | 4:28                 | 5:22                 |
>    | PCB Merging       | 3:13           | 4:27                 | 6:52                 |
>    | AdaMerging (k=1600)       | 11:28          | 27:37                | 52:13                |
>    | ProDistill (k=16)        | 27:09          | 29:23                | 49:20                |
>    | SAMerging (k=1600)         | 22:48          | 56:37                | 102:02               |

---

> > ### Author Response · Authors · 2025-11-26
> > **Response to Reviewer fHM9**
> >
> > **Q3: Merging on LLMs**
> >
> > We conducted experiments on merging LLMs for text classification. Note that our approach is for a classification task, as Theorem 3 puts a bound on the excess risk of a classifier. We used DeBERTa-v2-XXL, which is an **LLM with 1.5B parameters**. We take two fine-tuned versions of this LLM on MNLI and IMDb datasets for text classification. We also did the experiments on three more baselines: AdaMerging, TIES Merging, and Task Arithmetic. The result is as follows:
> >
> >
> > | Method          | MNLI   | IMDb   | Avg.    |
> > |-----------------|--------|--------|--------|
> > | Pre-trained     |  33.0  |  50.0  |  41.5  |
> > | Finetuned       |  91.7  |  97.1  |  94.4  |
> > | Task Arithmetic ($\lambda=0.5$) |  72.5  |  54.4  |  63.5  |
> > | Task Arithmetic ($\lambda=1.0$) |  **91.2**  |  65.9  |  78.6  |
> > | TIES-Merging |  88.7  |  *95.9*  |  92.3  |
> > | AdaMerging ($k=1600$)      |  72.5  |  54.4  |  63.5  |
> > | SAMerging ($k=1600$)      |   *90.5*  |  **96.4**  |  **93.5**  |
> >
> > As you can see, SAMerging outperformed baselines on LLMs for text classification.

---

### Official Review · Reviewer_kw7H · 2025-10-27

**Soundness:** 3
**Presentation:** 3
**Contribution:** 3
**Rating:** 6
**Confidence:** 3

**Summary:**

This paper proposes SAMerging, a method for model merging in the context of multi-task learning. SAMerging selects layer-wise merging coefficients by optimizing for flat minima and aligning with expert teachers via multi-teacher knowledge distillation on small amounts of unlabeled data. The theoretical backbone includes a PAC-Bayes generalization bound for the merged model, introducing a new cross-task heterogeneity term and connecting optimization of the merged model's sharpness and KL fit to rigorous generalization control. Empirically, SAMerging is tested on a range of computer vision and NLP benchmarks, consistently outperforming both data-free and data-dependent baselines, including AdaMerging, using significantly fewer calibration data, with no added inference overhead.

**Strengths:**

- This paper establishes a detailed PAC-Bayes generalization bound for MTL model merging and introduces an explicit cross-task heterogeneity term. This analysis provides motivation for practical design choices and clarifies failure modes.
- The method proposed in this paper addresses how to train a model with excellent performance even under zero initialization, which differs from previous works that require task arithmetic information.
- This paper also presents extremely detailed ablation experiments, which fully demonstrate the superiority of the SAMerging method.

**Weaknesses:**

- The paper does not include a comparison with the work ProDistill [1]. This is because the paper mentions that it requires 1,600 samples to achieve optimal performance, while the number of samples used in ProDistill is far fewer than that required by SAMerging. Is it necessary to further validate the conclusion that SAMerging requires fewer samples by comparing it with ProDistill?
- The description of experimental details is insufficient. The paper does not provide specific values for parameters such as ρ and η, nor does it conduct basic ablation experiments on these hyperparameters—these details need to be supplemented.
- The paper does not mention the memory overhead or time overhead during training. As the number of tasks increases, is the memory overhead completely proportional to the number of tasks? If so, how to address such significant memory overhead? If possible, please provide relevant calculation formulas or training methods.

[1] Jing Xu, Jiazheng Li, and Jingzhao Zhang. Scalable model merging with progressive layer-wise
distillation. (arXiv:2502.12706), May 2025. doi: 10.48550/arXiv.2502.12706. URL http:
//arxiv.org/abs/2502.12706. arXiv:2502.12706.

**Questions:**

My questions are listed with my weaknesses above.

I'm excited to engage with the authors to clear up the aspects I don't fully understand and I'm optimistic that with some iteration this paper can be made stronger.

---

> ### Author Response · Authors · 2025-11-22
> **Response to Reviewer kw7H**
>
> We sincerely thank the reviewer for his/her helpful comments. Below, we addressed the questions and will gladly incorporate the suggested improvements into the revised version to enhance the clarity of our work.
>
>
> **Q1: ProDistill Baseline**
>
> Thanks for pointing this out, we did the experiments to compare SAMerging with ProDistill. Note that ProDistill would take a long time running on $1600$ samples, so for a fair comparison, we tested both SAMerging and ProDistill on $16$ samples. Also, all other settings such as backbone, etc. are the same among the two methods. Here are the results:
>
> | Method | CLIP ViT-B/32 |  |  |  |  |  | CLIP ViT-L/14 |  |  |  |  |  |
> |--------|---------------|-------------|---------------|-------------|---------------|-------------|---------------|-------------|---------------|-------------|---------------|-------------|
> |        | **TA-8** |  | **TALL-14** |  | **TALL-20** |  | **TA-8** |  | **TALL-14** |  | **TALL-20** |  |
> |        | Acc. | Norm. | Acc. | Norm. | Acc. | Norm. | Acc. | Norm. | Acc. | Norm. | Acc. | Norm. |
> | Fine-tuned | 90.3 | 100.0 | 88.5 | 100.0 | 89.8 | 100.0 | 94.3 | 100.0 | 93.4 | 100.0 | 93.5 | 100.0 |
> | MTL | 88.5 | 98.0 | 87.7 | 99.1 | 88.9 | 99.0 | 92.3 | 97.9 | 91.6 | 98.1 | 91.8 | 98.2 |
> | ProDistill (k=16) | 81.1 | 89.8 | 80.5 | 91.0 | 77.8 | 86.6 | 87.2 | 92.5 | 89.0 | 95.3 | **86.8** | **92.8** |
> | **SAMerging (k=16)** | **83.8** | **92.8** | **81.2** | **91.8** | **77.9** | **85.7** | **91.2** | **96.7** | **89.1** | **95.4** | 85.8 |91.8 |
>
>
> We want to point out that ProDistill is learning an *elementwise* weight for each layer and task, so it is training a significantly larger model than SAMerging. (i.e., SAMerging is learning **a scalar for each layer and task**, while ProDistill is learning **a whole weight matrix for each layer and task**), but they do it in a progressive manner, allowing them to train a layer at a time and avoid the memory overhead of training a full model.
>
> **Q2:Hyperparameters and experiments**
>
> 2. Thanks for pointing this out. We added a section in Appendix D.1.3 to describe the experimental setup we follow for hyperparameters and ablations.
>
> 3. We would like to answer this question in three parts:
>
>    **(a) Memory overhead during merging:**
>
>    During merging, we have two overheads: (1) experts' models are loaded, and this increases linearly with the number of tasks, and (2) merging-dependent parameters. The second overhead depends on the merging method. For example, SAMerging is learning a scalar for each layer and task, while ProDistill is learning a whole weight matrix for each layer and task, progressively. Below is the memory overhead for each merging method for ViT-B/32 backbone on three task sets:
>    | Method        | TA-8 | TALL-14 | TALL-20 |
>    |---------------|-------------------|----------------------|----------------------|
>    | AdaMerging (k=1600)        | 4.2   GB          | 6.3 GB                | 8.3 GB                |
>    | ProDistill (k=16)        | **0.3 GB**          | **0.3 GB**                | **0.3 GB**                |
>    | SAMerging (k=1600)         | 8.3 GB          | 13.7 GB                | 19.2 GB               |
>
>     Note that our current implementation of SAMerging is not the most efficient one (we are keeping two sets of experts in memory, etc.); we'd expect to have a roughly similar memory footprint to AdaMerging if we implement it more efficiently.
>
>    **(b) Time overhead during merging:**
>
> Here is the time overhead for each merging method for ViT-B/32 backbone on TA-8: (Results are in (mm:ss) format, omit trivially simpler methods like Simple Averaging, Task Arithmetic, and TIES). However, note that if we want SAMerging to have the same *performance* as others, we need a shorter training time.
>    | Method        | TA-8 | TALL-14 | TALL-20 |
>    |---------------|-------------------|----------------------|----------------------|
>    | Fisher Merging    | 9:19           | 13:52                | 18:01                |
>    | RegMean           | 3:03           | 4:28                 | 5:22                 |
>    | PCB Merging       | 3:13           | 4:27                 | 6:52                 |
>    | AdaMerging (k=1600)       | 11:28          | 27:37                | 52:13                |
>    | ProDistill (k=16)        | 27:09          | 29:23                | 49:20                |
>    | SAMerging (k=1600)         | 22:48          | 56:37                | 102:02               |
>
>    **(c) Memory and Time overhead during inference:**
>    This factor goes back to our assumption that in the end of merging, we have a merged model that is no different than the jointly trained Multi-Task Model (which is trained on the data of all tasks). So the inference overhead is the same as each of the experts, basically we are running a model of same size than each of the experts, without any additional overhead in terms. We will include these results in revised version.

---

> > ### Comment · Reviewer_kw7H · 2025-11-26
> > **Official Comment by Reviewer kw7H**
> >
> > Thank you for your response. I've also reviewed the other reviewers' suggestions and decided to keep my positive rating.

---

### Official Review · Reviewer_P26t · 2025-10-30

**Soundness:** 2
**Presentation:** 3
**Contribution:** 3
**Rating:** 4
**Confidence:** 4

**Summary:**

This paper introduces SAMerging, a framework for data-efficient and label-free model merging. The authors derive a flatness-aware PAC-Bayes generalization bound that connects model sharpness with cross-task heterogeneity, providing theoretical insight into when merging succeeds. They further reinterpret coefficient learning as multi-teacher knowledge distillation, minimizing the KL divergence between the merged model and its experts while incorporating Sharpness-Aware Minimization (SAM) for better generalization. Extensive experiments on TA-8, TALL-14/20, and GLUE benchmarks show consistent competitive results.

**Strengths:**

1 SAMerging achieves consistent performance gains over both data-free and data-dependent baselines across multiple benchmarks.
2 The paper has a good organizational structure.

**Weaknesses:**

1 While the PAC-Bayes and SAM integration are new, the overall combination of KD + SAM resembles existing fine-tuning or merging extensions (e.g., AdaMerging + SAM). The contribution feels evolutionary rather than fundamentally new, since the method is essentially “AdaMerging + SAM + KD reformulation” without deeper empirical diversity.

2 The paper devotes extensive space to mathematical derivations (PAC-Bayes, NTK linearization, and multiple lemmas), but the empirical link between theory and practice is unclear. There is no ablation or visualization showing how the “flatness” or “heterogeneity” terms actually correlate with the final performance, making the theoretical results appear decorative rather than explanatory.

3 The experimental validation is confined to relatively standard benchmarks (TA-8, TALL-14/20, GLUE) using image classification and text classification tasks. This setting does not reflect the diversity or difficulty of modern model-merging scenarios. In particular,
all tasks share similar architectures and backbone initializations (CLIP or GPT-2), there are no tests on heterogeneous architectures, domain shifts, or large-scale multimodal models.

4 Several recent and competitive merging methods (e.g., Twin-Merging, PCB-Merging, 2024–2025) are missing, weakening the empirical thoroughness of the study.

5 Figure 2(a) is redundant, as its information is already presented in Table 1.

**Questions:**

Q1: Which previous does the experimental setup (e.g., Table 1) in this paper follow?

---

> ### Author Response · Authors · 2025-11-22
> **Response to Reviewer P26t**
>
> We sincerely thank the reviewer for his/her helpful comments. Below, we addressed the questions and will gladly incorporate the suggested improvements into the revised version to enhance the clarity of our work:
>
> **Q1: Contribution**
>
> Thanks for pointing this. We would like to point out our contribution in two aspects:
>    - To our knowledge, we are the first to derive a PAC-Bayes generalization bound that captures the complexity of the model-merging setting, beyond [1], which analyzes a simplified generalization and does not explicitly account for cross-task heterogeneity and the dependence on fine-tuned experts. In merging, we mix fine-tuned models, not data, which makes the analysis fundamentally different from standard single-model or joint multi-task generalization bounds [2, 3]. Because we do not have access to the underlying training data, we cannot rely on a standard empirical risk minimization framework on the merged model; instead, we must implicitly account for data heterogeneity *given* the expert models, leading to the cross-task heterogeneity term in our bound.
>    - The second aspect involved moving from empirical observations to establishing theoretical guarantees when selecting the loss. We take a step further compared to AdaMerging on why we should minimize KL for a theoretically-grounded bound of excess risk. This was not merely transitioning from entropy to KL but rather we *derived why we should minimize KL loss from scratch* which is interesting in its own.
>
> **Q2: Visualization**
>
> Thank you for this suggestion. We have added visualizations that directly connect flatness and task heterogeneity to performance. Specifically, we compare SAMerging (our method) with AdaMerging on TA-8, TALL-14, and TALL-20 (see Figure 3 in the PDF).
>
> In panels (a)–(c) in Figure 3, we plot the loss landscape around the merged model for pair of tasks (MNIST vs. DTD), where the axes are the scaling coefficients $\lambda$ on the corresponding task vectors. Two trends emerge:
>
> - Across all settings, the low-loss basins for SAMerging are wider than those for AdaMerging, indicating greater robustness to perturbations of the merged weights (look at the start point of the extreme red contour on the bottom right).
>
> - As we move from TA-8 to TALL-20 (more interference and heterogeneity), the safe low-loss region shrinks. This empirically supports the cross-task heterogeneity term in our bounds: increasing heterogeneity increases generalization loss.
>
> Panels (d)–(e) further analyze the EuroSAT vs. SUN397 pair by varying the scale of one task vector while tracking the loss of the other. The combined plot shows that SAMerging is more insensitive to these scaling changes than AdaMerging.
>
>  **Q3: Architecture**
>
>
> Thanks for pointing this out. We acknowledge that this is a limitation of our approach; however, we note that most existing methods similarly rely on this assumption. We consider it an interesting direction for future work. We will make this clear in the revised version.
>
>
> **Q4: Other baselines**
>
> We added three more baselines to the results, including ones you mentioned (For DistillPro results, refer to the response to reviewer kw7H). Note about Twin-Merging: In our assumption of model merging, we are seeking to find a parameter $\theta^\star$ from a set of parameters $\{\theta_i\}_{i=1}^T$ that can handle all the tasks without inference or memory overhead, and both will run in $\mathcal{O}(1)$. Twin-Merging violates this, incurring $\mathcal{O}(T)$ memory overhead and increased inference, making it a different category of method.
> Here are the results for more baselines (here we are showing the new baselines, for the other, refer to reviewr kw7H Q1):
>
> | Method | CLIP ViT-B/32 |  |  |  |  |  | CLIP ViT-L/14 |  |  |  |  |  |
> |--------|---------------|-------------|---------------|-------------|---------------|-------------|---------------|-------------|---------------|-------------|---------------|-------------|
> |        | **TA-8** |  | **TALL-14** |  | **TALL-20** |  | **TA-8** |  | **TALL-14** |  | **TALL-20** |  |
> |        | Acc. | Norm. | Acc. | Norm. | Acc. | Norm. | Acc. | Norm. | Acc. | Norm. | Acc. | Norm. |
> | Fine-tuned | 90.3 | 100.0 | 88.5 | 100.0 | 89.8 | 100.0 | 94.3 | 100.0 | 93.4 | 100.0 | 93.5 | 100.0 |
> | MTL | 88.5 | 98.0 | 87.7 | 99.1 | 88.9 | 99.0 | 92.3 | 97.9 | 91.6 | 98.1 | 91.8 | 98.2 |
> | **PCB Merging** | 75.4 | 83.5 | 70.3 | 79.4 | 64.1 | 71.4 | 84.2 | 89.3 | 80.4 | 86.1 | 72.6 | 77.6 |
> | **Twin Merging (see the note above)** | 90.4 | 100.1 | 89.3 | 100.9 | 89.8 | 100.0 | 90.4 | 95.9 | 93.4 | 100.0 | 93.5 | 100.0 |
> | **SAMerging** (k=1600) | 87.1 | 96.5 | 83.7 | 94.6 | 81.1 | 90.3 | 92.6 | 98.2 | 90.7 | 97.1 | 89.9 | 96.1 |
>
> **Q5: Redundancy of Figure 2 (a)**
>
> Thanks for pointing that out. We will replace the figure with the Q2 visualizations and add explanatory sections.

---

> > ### Author Response · Authors · 2025-11-22
> > **Response to Reviewer P26t**
> >
> > **Q1 in question: Experimental setup**
> > We follow the standard merging setup for CLIP and GPT (used by TIES, AdaMerging, and RegMean). We also added a subsection in Appendix D.1.3 to describe the experimental setup, such as hyperparameters for each method, in detail.
> >
> > References:
> > [1] Kim, T., Gouk, H., Kim, M., & Hospedales, T. (2025). Model merging is secretly certifiable: Non-vacuous generalisation bounds for low-shot learning. https://arxiv.org/abs/2505.15798
> >
> > [2] Zakerinia, H., & Lampert, C. H. (2025). Fast rate bounds for multi-task and meta-learning with different sample sizes. https://arxiv.org/abs/2505.15496
> >
> > [3] Alquier, P. (2024). User-friendly introduction to pac-bayes bounds. Foundations and Trends® in Machine Learning, 17(2), 174–303. https://doi.org/10.1561/2200000100

---

> > ### Comment · Reviewer_P26t · 2025-11-27
> > **This paper introduces SAMerging, a framework for data-efficient and label-free model merging.**
> >
> > Thank you for the authors' detailed rebuttal. While I appreciate the added visualizations and extra baselines, the core concerns on empirical breadth and the incremental nature of the method remain largely unresolved, and the theoretical connection is still not convincingly validated. I therefore keep my score.

---

> ### Author Response · Authors · 2025-11-28
> **Response to Reviewer P26t**
>
> Thank you very much for your time and thoughtful evaluation throughout the review process.
> We truly appreciate your careful consideration of our rebuttal. While we appreciate reviewer's perspective, we believe our approach takes steps to address the problems the reviewer mentioned for following reasons:
>
> 1. Theoretical Novelty and Incremental Nature: We introduce a novel PAC-Bayes generalization bound specifically derived for the model merging setting. Crucially, this bound **accounts for the fundamental shift from mixing data (standard learning) to mixing fine-tuned experts, capturing complexities like cross-task heterogeneity and lack of access to training data that standard bounds ignore**. Furthermore, unlike prior works (e.g., AdaMerging) that *empirically* adopt entropy, we theoretically derived the KL objective using principles of Multi-Teacher Knowledge Distillation. This derivation provides a proven bound on excess risk, distinguishing our framework as a principled design rather than an extension.
>
> 2. Empirical Breadth: We **visualized how seeking flatness will lead to better generalization and how increasing the task heterogeneity term will lead to harder generalization according to the derived bound**. Furthermore, to demonstrate versatility beyond standard benchmarks, we extended our evaluation to DeBERTa-v2-XXL (1.5B parameters) on MNLI and IMDb tasks. SAMerging achieved an average accuracy of 93.5%, outperforming competitive baselines like TIES (92.3%) and significantly surpassing AdaMerging (63.5%). This confirms that our theoretical findings hold for other types of models like LLMs.
>
> We are open to further discussion and addressing your concerns on these aspects of the work.

---

### Official Review · Reviewer_NAr3 · 2025-10-31

**Soundness:** 2
**Presentation:** 2
**Contribution:** 3
**Rating:** 4
**Confidence:** 2

**Summary:**

This paper derives a flatness-aware PAC-Bayes generalization bound that provides theoretical guidance for the design of model merging methods. The authors further propose SAMerging, which improves model merging through multi-teacher knowledge distillation on a small, unlabeled dataset. Theoretically, the paper proves that SAMerging tightens an upper bound on the merged model’s excess risk.

**Strengths:**

1. This paper theoretically identifies the key factors that influence model merging performance, providing guidance for the design of new merging methods.

2. The proposed method is also supported by solid theoretical analysis, which enhances the rigor and credibility of the approach.

3. The experimental results demonstrate that the proposed method achieves promising performance across various benchmarks.

**Weaknesses:**

1. **Unclear connection between the proposed method and the core Theorem 2.** The method introduced in Section 3.1 appears to have a weak connection with the main theoretical contribution presented in Theorem 2. In other words, it is unclear how the core theoretical result in Theorem 2 technically guides the design of the proposed SAMerging method in concrete technical details. Besides, Section 3.1 and Section 3.2 appear to follow two different theoretical frameworks, with limited connection between them.

2. **Reliance on training data.** The proposed approach depends on access to training data, which may limit its practical applicability, especially given the existence of several data-free model merging methods.

**Questions:**

See weaknesses above.

**Details Of Ethics Concerns:**

No ethics concerns.

---

> ### Author Response · Authors · 2025-11-22
> **Response to Reviewer NAr3**
>
> We sincerely thank the reviewer for his/her review. We address the raised weaknesses and questions below and will gladly incorporate them in the revised version to improve the clarity of our work regarding these concerns:
>
> **Q1: Unclear connection between the proposed method and the core Theorem 2.**
>
>
> We thank the reviewer for highlighting this point. In our analysis, Theorem 2 addresses *how* to perform the minimization, and Theorem 3 addresses *what* should be minimized. We clarify this connection in three steps:
>
>    1. In Section 3.1 and Theorem 2, we derived a generalization bound for the merged model. This generalization is for any convex loss $l$  in its score arguments, and not a specific loss. With that, we show that the generalization of the merged model is bounded by (1) flatness of the merged model $\sum_t \alpha_t\mathcal G_{\mathcal D_t}(\theta_{\mathrm{merge}}) + \sum_j \beta_j\mathcal G_{\mathcal D_j}(\theta_{\mathrm{merge}})$, and (2) flatness of the individual models $\mathcal{G}(\mathcal{D}(\mu_t))$. This implies that for a *given* loss $\ell$ that satisfies convexity in arguments, the generalization is bounded by the above factors, and hence, improving them leads to a better generalization. This suggests that a flatter minima ($\sum_t \alpha_t\mathcal G_{\mathcal D_t}(\theta_{\mathrm{merge}}) + \sum_j \beta_j\mathcal G_{\mathcal D_j}(\theta_{\mathrm{merge}})$) for the merged model will help the generalization of that. We show in step 3 that we operationalize this by leveraging SAM as the optimizer. This bound also justifies the improvement observed by methods like SAFT [3], which aims at minimizing $\mathcal{G}_\mathcal{D}(\mu_t)$.
>
>
>    2. So far in step 1, we talked about an arbitrary loss $\ell$; in Section 3.2, we specify that loss. Unlike AdaMerging, which empirically showed that entropy loss is a desirable objective, we are aiming to theoretically derive this loss. For example, entropy loss has a global minimum (e.g., a model that, regardless of input, classifies it as class 1), which is obviously undesired, and avoiding this needs assumptions on the quality of the pretrained model, which is out of the merging scope. That is why we aim to construct the framework for the excess risk of classification to derive an objective that controls the excess risk of the merged model. To that end, we frame the model merging as multi-teacher knowledge distillation, as in Theorem 3. Our objective is not a mere change from entropy to KL, but it is rooted in how the merged model, as a student, would learn a task from its teachers/experts. This suggests that if we use $\mathrm{KL}$ as the objective, we can theoretically bound the merged model classification excess risk without needing the label of the data. This bound also explains the reliance of the merged model on the quality of experts.
>
>
>    3. In Section 3.3, we operationalize our findings from 3.1 and 3.2. SAMerging minimizes the KL between the merged model and experts as the objective in a way to promote the flatness of the merged model.
>
>
> We hope this clarifies the relationship, and we will make it explicit in the revised version. Please also see response to Q1 of reviewer fHM9 on contribution.
>
> **Q2: Reliance on training data.**
>
> Thanks for pointing this out. First, we would like to point out that there are different assumptions on accessing data in model merging methods. We would like to briefly point out to them as follows:
>
>    1. Accessing training data: Methods like RegMean, assume we have access to training data or its Gram.
>    2. Accessing test data: methods like SAMerging and AdaMerging [1] do not have access to *training* data, which is a typical assumption in model merging literature. Instead, **we are relying on limited unlabeled test data** (i.e., test-time adaptation).
>
>    Note that in both cases, we are assuming we don't have access to labels. Now with that in mind, there are methods that do not use this data at all, which we referred to as data-free methods. Two arguments drive us to accept the reasonable assumptions at the cost of relying on a **small subset of unlabeled data in test-time**:
>
>    1. There are works (e.g., Theorem 2 in [2]) that show data-free merging methods can be **arbitrarily bad**. In other words, one can always have two tasks for which the data-free merging methods can work arbitrarily bad.
>    2. As shown in the experiments, the performance of data-free methods is usually much weaker than data-dependent methods. We can see that **SAMerging with only 16 unlabeled examples: $83.8\%$ vs. Isotropic Merging: $78.8\%$**.
>
> These considerations led us to conclude that, in realistic settings, leveraging data at test time can yield better performance.
>
> References:
> [1] Yang, E. et al. Adamerging arxiv.org/abs/2310.02575
>
> [2] Xu, J., Scalable model merging arxiv.org/abs/2502.12706
>
> [3] Lee, Y. et al. Mitigating parameter interference in model merging via sharpness-aware fine-tuning. arxiv.org/abs/2504.14662

---

### Author Response · Authors · 2025-11-23
**Summary of Revision and Invitation for Further Discussion**

We thank all reviewers for their constructive and insightful feedback. In response, we have  clarified the paper's contributions, strengthened the empirical evaluation with additional baselines, and improved the presentation.

**Clarified contributions and theory.**

We now state our main contributions more explicitly in the introduction and conclusion, and better explain how the theoretical results guide the design of SAMerging. In particular, we clarify the roles of the key theorems: one explains *how* generalization can be controlled via flatness terms for an arbitrary convex loss, and another specifies why minimizing a KL-based objective is the appropriate choice in the model merging setting. We also add brief intuitive explanations before technical statements so that the main ideas are easier to follow.

**Additional baselines and experimental details.**

We expanded the empirical section with new baselines and clearer comparisons. We now include results for methods such as ProDistill, PCB Merging, and Twin Merging under matched settings (same backbones, same heterogeneity, etc.). We also make explicit why some approaches, such as Twin Merging, do not match our constant-overhead constraint due to their $\mathcal{O}(T)$ memory or inference cost. The experimental setup is described in more detail, including training hyperparameters and ablations, so that our comparisons are reproducible, and we have put it in Appendix D.1.3. We also extended the merging to LLMs with 1.5B parameters, showing that SAMerging outperforms baselines on large models. (Table 1, Table 3, and Table 12 of the paper, Q1 of kw7H, Q4 of P26t)

**NTK assumptions and empirical support.**

To support the NTK assumptions underlying our analysis, we add diagnostics that quantify how far experts and merged models move from the pretrained initialization. We report Frobenius norms and show that models typically stay in a small neighborhood around initialization, with larger backbones moving even less. We also show that SAMerging often produces merged models closer to the pretrained model than competing methods like AdaMerging, aligning with the regime assumed in our theoretical development. We add these details to the paper to explain the empirical link to the assumptions. (Q2, reviewer fHM9)

**New visualizations and interpretation.**

We revise the figures to better illustrate the connection between theory and practice. We now plot the loss landscape for pairs of tasks around the merged model in different interference settings. New visualizations highlight (i) how SAMerging finds flatter low-loss regions compared to other merging methods and (ii) how increasing cross-task heterogeneity shrinks the safe region, matching the behavior predicted by our heterogeneity term. These plots help readers see how loss of flatness and task heterogeneity affect performance (Q2, reviewer P26t and Figure 3 of the paper).

**Presentation and organization improvements.**

We improve the overall structure of the paper by adding new visualizations and clarifying the presentation of the paper.

We believe these revisions address the main concerns raised in the reviews and make the paper's contributions, empirical evidence, and practical implications clearer. We welcome any further questions or insights to refine our work.

---

### Comment · Area_Chair_tsEk · 2025-11-27
**Request for Timely Response to Authors’ Rebuttal and Discussion**

Dear Reviewers,

I hope you are doing well. The authors have now submitted their rebuttal for the paper under your review. At this stage, your timely response is essential for ensuring a smooth discussion phase.

Could you please review the rebuttal at your earliest convenience and share your updated thoughts? If there are points that require further discussion among the reviewers, please feel free to initiate or join the conversation on the discussion thread.

Your prompt input will greatly help us maintain the review timeline. Thank you very much for your efforts and valuable contributions.

Best regards,

AC

---

### Meta-Review · Area_Chair_5Mki · 2026-01-05

**Summary:**

This paper introduces SAMerging, a model merging framework that integrates PAC-Bayes–motivated KL-based multi-teacher distillation with sharpness-aware minimization to improve multi-task performance. The paper provides theoretical analyses attempting to justify the link between sharpness, cross-task heterogeneity, and generalization, and presents empirical results on vision and NLP benchmarks.

While the reviewers acknowledge the novelty of applying PAC-Bayes theory to multi-task model merging and appreciate the clear organizational structure and ablation studies, several substantial concerns remain insufficiently resolved after rebuttal:

1. **Incremental conceptual contribution**:

Both reviewer P26t and fHM9 expressed concern that the method is conceptually close to existing approaches (e.g., AdaMerging + entropy minimization + SAM), with the novelty appearing incremental rather than foundational. The rebuttal does not sufficiently address this perception, as the paper still lacks strong evidence that the theoretical terms directly lead to algorithmic advantages beyond heuristic motivation.

2. **Unclear theoretical–algorithmic connection**:

Reviewer NAr3, P26t and fHM9 highlighted that the theoretical results do not clearly justify the specific algorithmic design choices. In particular, it remains unclear why the "flatness" is correlated with the final performance (reviewers NAr3, P26t), and to what extent the NTK-based assumptions meaningfully support generalization claims in practical nonlinear regimes (reviewer fHM9).

3. **Substantial time and memory overhead**:

A key remaining concern is the substantial time and memory overhead of SAMerging during merging (reviewer kw7H), which remains significantly higher than baselines. Given the cost, the practical applicability of the method in realistic multi-task or large-scale settings remains questionable despite added clarifications.

Given these concerns, the current theoretical and empirical support is insufficient for acceptance. I recommend rejecting the paper at this stage and encourage the authors to strengthen the theory–algorithm connection and expand experiments before resubmission.

**Reviewer Concerns:**

**Addressed by the rebuttal**:

- Missing baselines (PCB-Merging, Twin-Merging, ProDistill) have been added, resolving empirical-diversity concerns (reviewers P26t, kw7H).
- Lack of LLM evidence was partially addressed by adding DeBERTa-v2-XXL experiments (reviewer fHM9).

**unsolved problems**:

- Incremental-nature concern regarding novelty remains largely unaddressed, as the method is still perceived as combining existing components rather than introducing a fundamentally new paradigm (reviewers P26t, fHM9).
- The theory and algorithm connection remains only partially justified; the PAC-Bayes derivation still does not clearly compel the chosen KL objective or SAM design (reviewers NAr3, P26t, fHM9).
- The issue of high time and memory cost during merging remains unresolved despite explanations (reviewer kw7H).

**Reviewer Scores:**

- **Reviewer NAr3**: While the rebuttal clarified the data reliance question, it did not substantially resolve the core theoretical–algorithmic connection concerns. Thus, I expect this reviewer would likely maintain the original score.
- **Reviewer P26t**: After the author's rebuttal, the reviewer mentioned that the incremental nature concern and theoretical connection are still outstanding, thus the reviewer did not change the score.
- **Reviewer kw7H**: After the author's rebuttal, the reviewer maintained the positive score.
- **Reviewer fHM9**: Although the rebuttal clarified the NTK practicality problem and added LLM experiments, the major concerns regarding the incremental nature and theoretical–algorithmic connection are still present. Thus, I expect this reviewer would likely maintain the original score.

---

### Decision · Program_Chairs · 2026-01-26

Reject